# Expression of CD20 after viral reactivation renders HIV-reservoir cells susceptible to Rituximab

Carla Serra-Peinado [1], Judith Grau-Expósito [1], Laura Luque-Ballesteros [1], Antonio Astorga-Gamaza[1], Jordi Navarro [1], Jenny Gallego-Rodriguez[1], Mario Martin [1], Adrià Curran [1], Joaquin Burgos [1], Esteban Ribera [1], Berta Raventós [1], Rein Willekens [1], Ariadna Torrella [1], Bibiana Planas [1], Rosa Badía [1], Felipe Garcia [2], Josep Castellví [3], Meritxell Genescà[1], Vicenç Falcó [1] & Maria J. Buzon [1]

The identification of exclusive markers to target HIV-reservoir cells will represent a significant advance in the search for therapies to cure HIV. Here, we identify the B lymphocyte antigen CD20 as a marker for HIV-infected cells in vitro and in vivo. The CD20 molecule is dimly expressed in a subpopulation of CD4-positive (CD4$^+$) T lymphocytes from blood, with high levels of cell activation and heterogeneous memory phenotypes. In lymph node samples from infected patients, CD20 is present in productively HIV-infected cells, and ex vivo viral infection selectively upregulates the expression of CD20 during early infection. In samples from patients on antiretroviral therapy (ART) this subpopulation is significantly enriched in HIV transcripts, and the anti-CD20 monoclonal antibody Rituximab induces cell killing, which reduces the pool of HIV-expressing cells when combined with latency reversal agents. We provide a tool for targeting this active HIV-reservoir after viral reactivation in patients while on ART.

[1] Infectious Disease Department, Hospital Universitari Vall d'Hebrón, Institut de Recerca (VHIR), Universitat Autònoma de Barcelona, Barcelona, Spain. [2] Infectious Disease Department, Hospital Clínic de Barcelona, Universitat de Barcelona, Barcelona, Spain. [3] Department of Pathology, Hospital Vall d'Hebrón, Universitat Autònoma de Barcelona, Barcelona, Spain. Correspondence and requests for materials should be addressed to M.J.B. (email: mariajose.buzon@vhir.org)

HIV infection represents an incurable disease to date. The persistence of the HIV-reservoir that is not susceptible to current antiretroviral therapy (ART) represents one of the main barriers to cure HIV[1,2].

During the last two decades, intense efforts have focused on the search for new markers to successfully identify latently HIV-infected cells; cells that contain integrated HIV genetically intact but able to remain quiescent for prolonged periods in vivo. Recently, the paradigm that most latent HIV is transcriptionally inactive has been questioned by a new investigation that showed that most HIV-infected CD4-positive (CD4$^+$) T cells were able to successfully initiate HIV transcription[3]. During ART, reservoir cells expressing HIV RNA are ~10 times more frequent than the proportion of cells containing replication-competent HIV and are found in diverse populations of infected cells, including physiologically activated cells that produce viral particles upon cell stimulation[4,5]. HIV-infected cells might also become pharmacologically activated to produce HIV transcripts by latency reversal agents (LRAs), which are compounds that are currently being assessed in clinical trials for their ability to reactivate HIV from its dormant state. LRAs have been shown to be effective in inducing HIV transcription, but importantly, viral-reactivated cells are not completely cleared by the immune system or killed by viral cytopathic effects in vivo[6–8]. Thus, the identification of novel markers to identify transcriptionally active HIV-infected cells, including physiologically and/or pharmacologically HIV-reactivated cells, could considerably advance the development of novel therapies to target and deplete the HIV-reservoir.

CD20 is a cell membrane receptor preferentially expressed on B cells. Early reports have also demonstrated the existence of a proportion of CD3-positive (CD3$^+$) T cells with dim expression levels of CD20 (between 1 and 5% in blood)[9–12]. In healthy individuals, CD20-positive (CD20$^+$) T cells are phenotypically different than their counterpart CD20-negative (CD20$^-$) T cells; approximately half of them are CD4$^+$ cells, they are more likely to express the gamma/delta T-cell receptor, and they express more CD45RO and fewer CD38 molecules on their surface[12]. In patients suffering from rheumatoid arthritis[13,14] CD20$^+$ T cells have been found to have a low proliferative capacity, a high activation state, and an enhanced susceptibility to apoptosis[9]. In patients with multiple sclerosis, in vivo administration of the anti-CD20 monoclonal antibody Rituximab effectively induces long-term depletion of T cells expressing CD20[11]. Moreover, CD20$^+$ T cells have been postulated as early T-cell precursors in the replenishment of diverse populations of T cells[11,12]. Therefore, the intrinsic characteristics of CD20$^+$ T cells and their susceptibility to Rituximab administration in vivo make the CD20 molecule a good candidate to explore in the setting of HIV reservoirs. Here, we investigate CD20 as a putative marker for HIV infection and during in vivo viral persistence. We find that CD20 expression in CD4$^+$ T cells is expanded during untreated HIV infection, and during ART, CD20$^{dim}$ T cells contain high levels of HIV transcripts. Importantly, Rituximab induces cell death and decreases the size of the transcriptionally active HIV-reservoir when combined with LRAs ex vivo. Thus, we identify CD20 as a new marker for HIV-infected cells in patients and provide evidence for the combined use of LRAs and anti-CD20 monoclonal antibody therapy for the depletion of HIV-reservoir cells after viral reactivation.

## Results

### A subset of CD4$^+$ T cells expresses CD20 in HIV infection.
First, we aimed to determine the proportion of CD4$^+$ T cells expressing the CD20 molecule during HIV infection in samples from 21 ART-suppressed patients (<50 copies/ml of plasma viral load, VL; median time on suppressive ART 48 months), 20 HIV-viremic patients (median VL 119,000 copies/ml) and 6 uninfected control donors. The characteristics of individual patients are shown in Supplementary Table 1. The population of CD4$^+$ T cells expressing CD20 was identified according to the fluorescence minus one (FMO) control for CD20 expression (Fig. 1a). As shown in Fig. 1a and in agreement with previous studies[12], we found that CD4$^+$ T cells expressed dim levels of the CD20 receptor (CD20$^{dim}$). Importantly, the inclusion of the CD19 marker to exclude B cells did not alter the frequency of CD20$^{dim}$ CD4$^+$ T cells (Supplementary Fig. 1a). To define the lymphocyte lineage of CD20$^{dim}$ CD4$^+$ T cells, we performed fluorescent flow cytometry imaging (Amnis®) on samples from HIV-infected patients. When cells were gated based on the expression of CD3, CD4 and dim levels of CD20, we observed the unequivocal expression of CD20 uniformly distributed through the plasma membrane of CD4$^+$ T cells. Lack of expression of CD19 on these CD20$^{dim}$ cells corroborated that they did not belong to a canonical B-cell phenotype (Fig. 1b). B cells were highly positive for CD19 and CD20 expression (Fig. 1c). Of note, we also found CD4$^+$ T cells with high expression of CD20, however, the flow cytometry imaging analysis showed that they corresponded to cell aggregates consisting of T-B-cell doublets (Supplementary Fig. 1b). In addition, conventional quantitative PCR showed the dim expression of CD20-mRNA in previously isolated CD20$^{dim}$ CD4$^+$ T cells (Supplementary Fig. 1c). Next, we quantified the percentage of CD20 in CD4$^+$ T cells of HIV$^+$ patients and uninfected controls. We observed that viremic HIV$^+$ patients presented statistically significant higher proportions of CD4$^+$ T cells expressing CD20 compared with uninfected donors (Fig. 1d). Moreover, CD4$^+$ T cells from HIV$^+$ viremic individuals had slightly higher expression of CD20 than virologically suppressed HIV$^+$ patients. The antiretroviral treatment normalized the expression of CD20 since no differences were found between ART-suppressed patients and uninfected controls (Fig. 1d). When we compared the activation levels, defined by the expression of HLA-DR, CD69, CD25 or any combination of these cell markers, we observed that in HIV$^+$ patients CD20$^{dim}$ CD4$^+$ T cells were more activated than CD4$^+$ T cells not expressing CD20 (Fig. 1e). Next, we studied the memory phenotype of these cells in HIV infection. In terms of the distribution of CD20$^{dim}$ among the different CD4$^+$ T cell subsets, most of the CD20$^{dim}$ cells had a central memory (T$_{CM}$) phenotype, followed by the effector memory (T$_{EM}$), transitional memory (T$_{TM}$), and naive (T$_{NA}$) phenotypes. CD20$^{dim}$ cells were less distributed among memory stem cells (T$_{SCM}$) and terminally differentiated (T$_{TD}$) cells (Fig. 1f). Thus, all the memory subsets accounted for almost 80% of the total expression of CD20$^{dim}$ CD4$^+$ T cells. Simultaneously, we observed that memory subsets, including CD4$^+$ T$_{CM}$, CD4$^+$ T$_{EM}$, and CD4$^+$ T$_{TM}$ cells, contained significantly higher percentages of CD20$^{dim}$-expressing cells, as compared with the less differentiated CD4$^+$ T$_{NA}$ and CD4$^+$ T$_{SCM}$, or with the CD4$^+$ T$_{TD}$ cells (Supplementary Fig. 2a). In this regard, small differences in CD4$^+$ T cell subsets were observed between ART-suppressed individuals, HIV-viremic individuals, and uninfected controls (Supplementary Fig. 2b). When we analyzed all CD4$^+$ T memory cells together, defined by the expression of CD45RO, we found the same trend observed in the total CD4$^+$ T-cell population (shown in Fig. 1f), suggesting that these differences are mainly supported by the memory pool (Supplementary Fig. 2c). We also studied if CD20$^{dim}$ CD4$^+$ T cells expressed cell markers characteristics of follicular T helper (T$_{FH}$) cells, defined by the expression of PD1 and CXCR5. We observed that CD20$^{dim}$ cells expressed significantly more double positive CXCR5$^+$ and PD1$^+$ than CD20$^-$ CD4$^+$ T cells (Supplementary Fig. 1d).

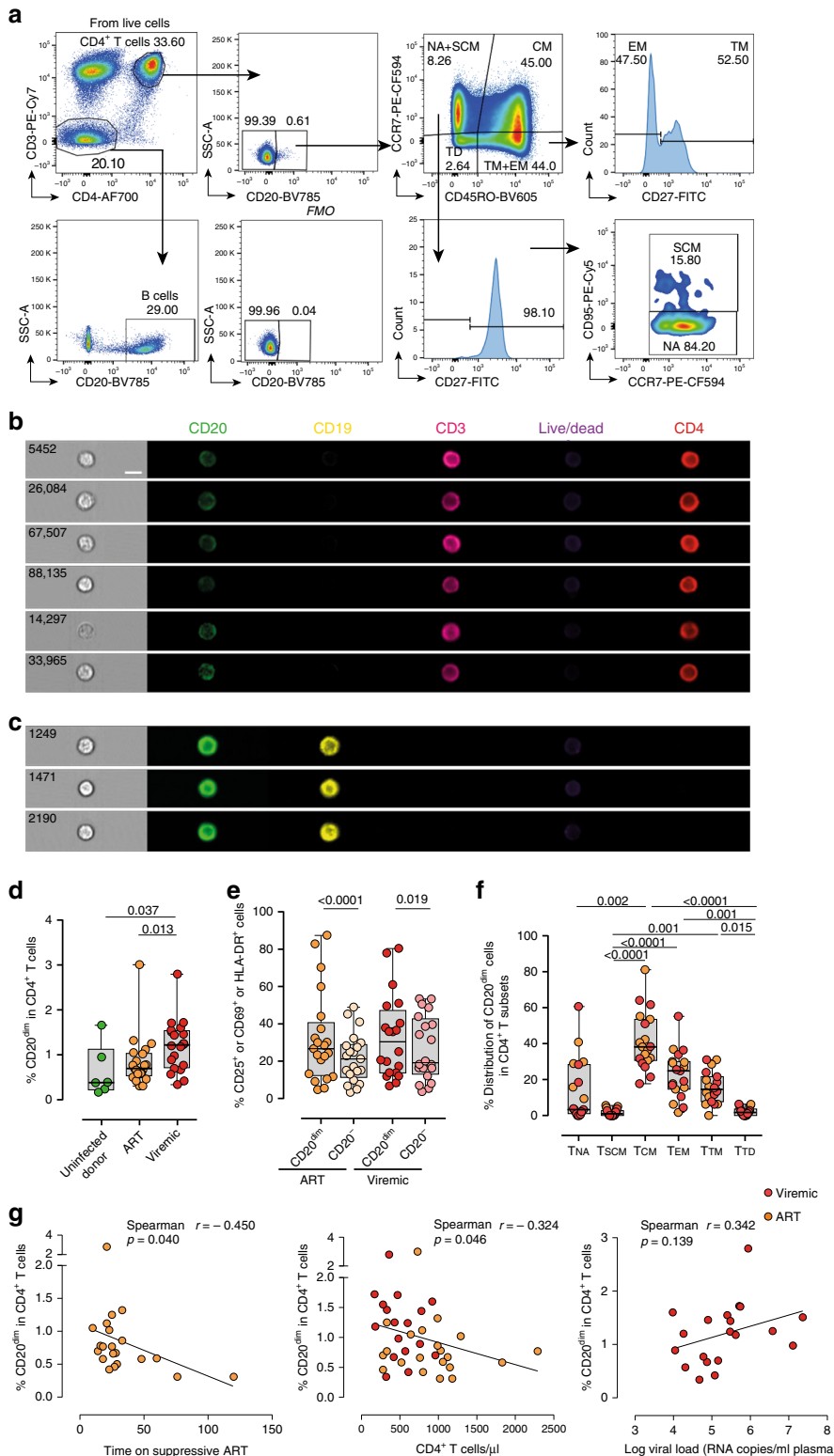

Finally, we evaluated the relationship between the expression of CD20 in T cells and markers of HIV progression. A significant inverse correlation was observed between the percentage of CD20$^{dim}$ CD4$^+$ T cells and the number of months that patients were under suppressive treatment (VL < 50 copies/ml), and with their CD4$^+$ T-cell count (Fig. 1g). A tendency toward a positive correlation was also observed with the plasma viral load (Fig. 1g). In summary, HIV$^+$ individuals had higher expression of CD20 on

CD4$^+$ T cells compared with uninfected controls that inversely correlated with CD4$^+$ T cell counts and time on suppressive ART. Moreover, the CD20 marker was associated with activated cells and mainly expressed and distributed in memory CD4$^+$ T-cell subsets.

**CD20$^{dim}$ CD4$^+$ T cells are enriched in intracellular HIV RNA.** Since CD4$^+$ T cells expressing the marker CD20 seem to be more

**Fig. 1** CD20$^{dim}$ CD4$^+$ T cells involve memory phenotypes with more activation than CD20$^-$ cells. PBMC from uninfected donors, ART-suppressed patients and viremic patients were stained with CD20, activation markers (HLA-DR, CD69, CD25), and T-cell differentiation markers (CCR7, CD45RO, CD27, CD95). **a** Gating strategy used to identify CD20$^{dim}$ CD4$^+$ T cells. Previous sequential gates are represented in Supplementary Fig. 9. Viable CD4$^+$ T cells (identified by CD3 and CD4 expression) were first selected for the analysis of CD20 expression, defined by its *Fluorescence Minus One* (FMO) control. T-cell memory subsets were selected as follows: CD4$^+$ T$_{NA}$ (CCR7$^+$, CD45RO$^-$, CD27$^+$, CD95$^-$), CD4$^+$ T$_{SCM}$ (CCR7$^+$, CD45RO$^-$, CD27$^+$, CD95$^+$); CD4$^+$ T$_{CM}$ (CCR7$^+$, CD45RO$^+$), CD4$^+$ T$_{TM}$ (CCR7$^-$, CD45RO$^+$, CD27$^+$); CD4$^+$ T$_{EM}$ (CCR7$^-$, CD45RO$^+$, CD27$^-$), CD4$^+$ T$_{TD}$ (CCR7$^-$, CD45RO$^-$). **b, c** Representative bright-field and pseudo-color fluorescence images of CD20$^{dim}$ CD4$^+$ T cells (**b**) and B cells (**c**) from two ART-suppressed patients (#9 and #22) using the Amnis imaging flow cytometer technology. Scale bar 10 μm. **d** Percentage of CD20$^{dim}$ expression within CD4$^+$ T cells in uninfected controls and the two patient cohorts. Mann–Whitney comparison was used to compare $n = 6$ uninfected controls, $n = 21$ ART-suppressed patients, $n = 20$ viremic patients. **e** Expression of the activation markers CD25, CD69 or HLA-DR in CD20$^{dim}$ and CD20$^-$ CD4$^+$ T cells in different cohorts of HIV$^+$ patients. Comparisons were performed using the two-tailed Wilcoxon test ($n = 21$ ART-suppressed patients and $n = 20$ viremic patients). **f** Distribution of CD20$^{dim}$ in CD4$^+$ T-cell subsets in ART-suppressed (orange) and viremic (red) HIV-infected patients. NA, naive; SCM, memory stem cell; CM, central memory; TM, transitional memory; TD, terminal differentiated. ANOVA and Dunn's multiple comparison test were performed, $n = 10$ ART-suppressed and $n = 9$ viremic patients. Median values and min and max ranks are represented in panels **d–f**. **g** Correlation of CD20$^{dim}$ CD4$^+$ T cells with time on suppressive ART (left), CD4$^+$ T-cell counts (middle) and plasma viral load (right) are shown. ART-suppressed patients in orange and viremic patients in red. Spearman's nonparametric correlation coefficients and associated *P* values are shown. Panels (**d**) and (**e**) included patients #2–10, 15–26, 60–65, 67–74, and 76–81. Panel (**f**) patients #1–10, 60–65, and 67–69. Panel (**g**) patients #2–10, 15–26, 60–65, 67–74, and 76–81. Data underlying this Figure are provided as Source Data file

activated than the general population of CD4$^+$ T cells, and cell activation has been related to higher levels of HIV-1 infection and transcription, we sought to investigate if CD20$^{dim}$ CD4$^+$ T cells from HIV$^+$ patients might also contain more transcriptionally active HIV. In general, no correlation was found between the expression of CD20 in CD4$^+$ T cells and viral nucleic acids during ART (Supplementary Fig. 2d). Using the novel RNA FISH-flow assay[4], we measured the frequency of CD20$^{dim}$ CD4$^+$ T cells that were co-expressing HIV RNA. We included samples from 12 ART-suppressed patients (median time on suppressive ART 23 months), 11 viremic patients (median VL 74,500 copies/ml) and 4 uninfected controls. The characteristics of the included patients are shown in Supplementary Table 1. The gating strategy used to identify CD20$^{dim}$ CD4$^+$ T cells and HIV RNA expression is shown in Fig. 2a. Both cohorts of HIV-infected patients, viremic and ART-suppressed patients, presented a significantly higher proportion of CD20$^{dim}$ CD4$^+$ T expressing viral transcripts compared with their counterpart CD20$^-$ CD4$^+$ T cells (Fig. 2b). These infected CD20$^{dim}$ CD4$^+$ T cells contributed to a median of 18.55% and 25.0% to the total pool of HIV-expressing cells in ART-suppressed and viremic patients, respectively (Fig. 2c). Overall, we observed that CD20$^{dim}$ CD4$^+$ T are enriched in HIV RNA and contribute significantly to the total pool of HIV-expressing cells.

**Proviral HIV DNA levels of CD20$^{dim}$ CD4$^+$ T cells.** We next asked if the increased viral transcription observed within the CD20$^{dim}$ CD4$^+$ T population was associated with enrichment in viral DNA. We used fluorescence activated cell sorting to isolate total CD4$^+$ T cells, activated CD4$^+$ T cells (CD20$^-$), resting CD4$^+$ T cells (CD20$^-$), activated CD20$^{dim}$ CD4$^+$ T cells, and resting CD20$^{dim}$ CD4$^+$ T cells from nine HIV-infected ART-suppressed individuals (median time on suppressive ART of 156 months) (Supplementary Table 1). Resting CD20$^{dim}$ CD4$^+$ T cells had significantly higher levels of HIV DNA compared with the total CD4$^+$ T cells (Fig. 3a, left panel). This enrichment represented a 2.1-fold change in the amount of HIV DNA per million cells between both subsets (Fig. 3a, right panel). Moreover, we calculated the global contribution of these subsets to the total pool of HIV-infected cells, and we found that resting CD20$^{dim}$ contributed to a median of 13.26% compared with 25.44% and 57.79% of activated CD4$^+$ T cells and resting CD4$^+$ T cells, respectively (Fig. 3b).

**CD20$^+$ T cells are infected in human lymph nodes.** We next visualized and quantified the CD20 receptor in anatomically intact lymph node tissue preparations from two ART-suppressed patients and four viremic HIV-infected patients (Supplementary Table 1). First, we focused on the quantification of T cells co-expressing the marker CD20. We detected CD20$^+$ T cells in both the extrafollicular zone and within the B-cell follicle in all samples analyzed. Representative micrographs are shown in Fig. 4a. In the sample from one HIV-viremic patient (#82), we observed significantly higher absolute number of CD20$^+$ T cells within the B-cell follicle compared with the extrafollicular region, while the two ART-suppressed patients had lower quantities of CD20$^+$ T cells in B-cell follicles compared with the viremic patients (Fig. 4b, upper panel). Also, we quantified the percentage of follicular CD3$^+$ cells that co-expressed CD20. We found that, in general, the median percentage of CD3$^+$ cells positive for CD20 expression ranged from 2 to 4% in individual patients (Fig. 4b, lower panel). Next, we investigated whether HIV-infected cells were also expressing CD20 in lymph node tissue specimens from three viremic and one ART-suppressed HIV-infected patients. Since productively HIV-infected cells have been previously identified as being CD45RO-positive memory CD4$^+$ T cells[15], we did not perform phenotypic identification of such cells. Instead, we directly performed immunostaining of the viral protein p24 and the CD20 antigen. P24 showed the typical two-staining pattern observed in the B-cell follicles[16]; a network-like staining probably corresponding to virions captured by follicular dendritic cells (FDCs), and cells with densely spherical signal indicating productively HIV-infected cells. Notably, within the B-cell follicles of all patients, we observed productively HIV-infected cells co-expressing the CD20 molecule (Fig. 4c), but cell quantification was difficult due to the high density of B cells expressing CD20. To finally demonstrate the expression of CD20 in productively HIV-infected cells, we used the in situ hybridization ultrasensitive RNA detection assay (eBiosciences) to visualize HIV RNA and CD20 RNA in a lymph node section of one HIV$^+$ viremic patient. In this particular patient, productively HIV-infected cells were often associated with the expression of CD20, inside and outside of the B-cell follicle (Fig. 4d). Overall, we confirm the existence of HIV-infected cells expressing CD20 in lymph nodes from ART-suppressed and viremic HIV-infected patients.

**HIV upregulates CD20 in T cells during early viral infection.** To investigate the role of HIV-1 replication in the expression of

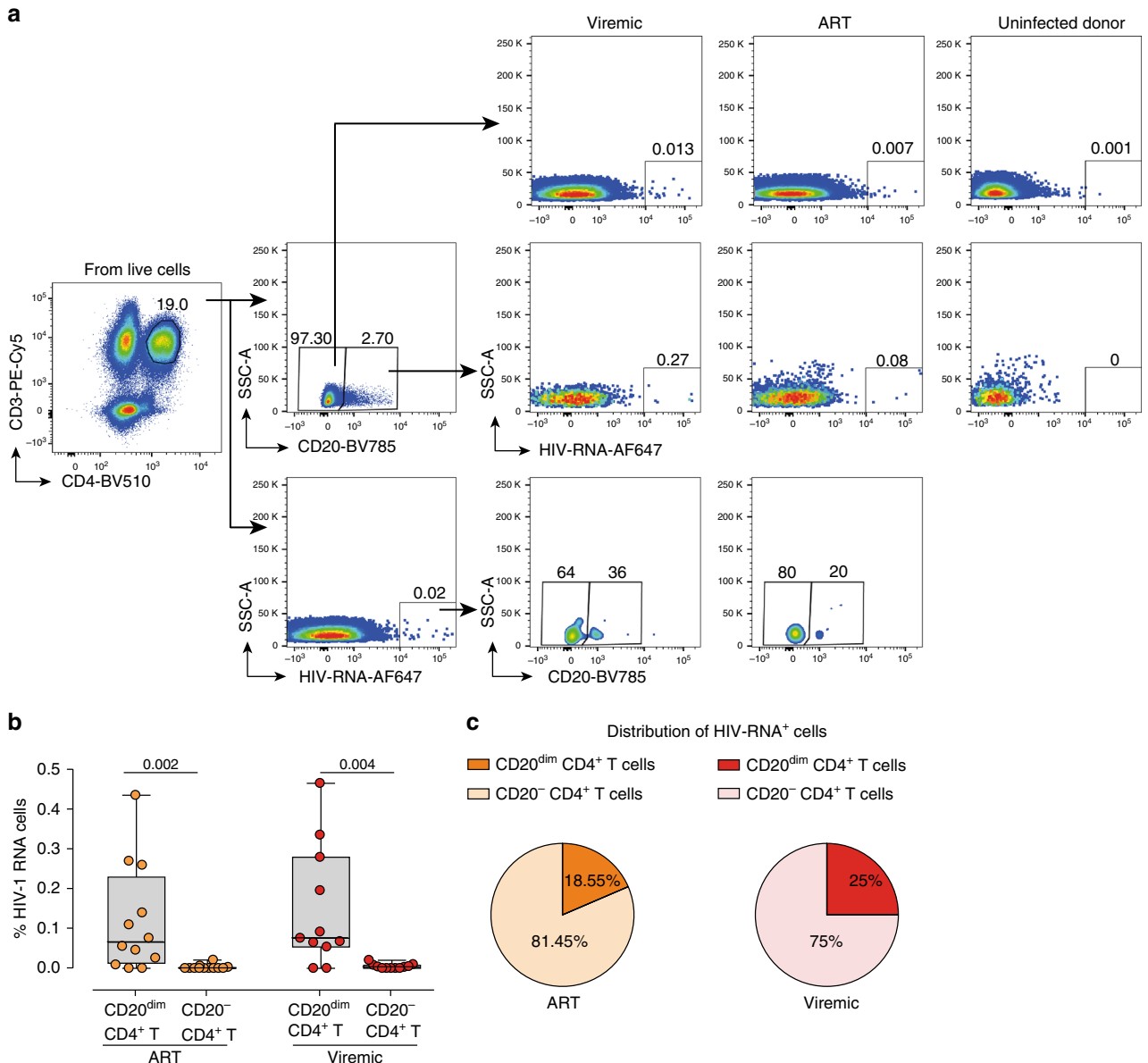

**Fig. 2** CD20$^{dim}$ CD4$^+$ T cells are enriched in HIV-1 RNA. Unstimulated PBMCs were subjected to the RNA FISH-flow protocol to detect HIV RNA expression in CD20$^{dim}$ CD4$^+$ T cells. **a** Gating strategy used to determine the HIV RNA expression on CD20$^-$ and CD20$^{dim}$ CD4$^+$ T cells and expression of CD20 in HIV RNA$^+$ cells. Previous sequential gates are represented in Supplementary Fig. 9. **b** Frequency comparison of HIV RNA$^+$ cells in CD20$^{dim}$ CD4$^+$ T cells and CD20$^-$ CD4$^+$ T cells of ART-suppressed and viremic patients. Wilcoxon test was used. **c** Mean distribution of HIV RNA$^+$ cells in the CD20$^+$ and CD20$^-$ phenotypes. Median values and min and max ranks are represented in panels (**b**) and (**c**). In all panels, $n = 12$ ART-suppressed and $n = 11$ viremic patients are shown. Data from patients #15–26, 60, 63, and 73–81 are included. Data underlying this Figure are provided as Source Data file

CD20, we performed ex vivo infection experiments. Unstimulated PBMCs from three uninfected donors were infected with the viral strain NL4.3 for 5 days. The gating strategy for productively HIV-infected cells, identified as HIV RNA-positive (HIV RNA$^+$) and p24-positive (p24$^+$) T cells, and expression of CD20 are shown in Fig. 5a. Ex vivo infection was successfully detected using this novel assay; we obtained increasing percentages of HIV infection over time (Fig. 5b left). Interestingly, when we focused on CD20$^{dim}$ T cells, we observed that this subpopulation supported HIV infection more efficiently than CD20$^-$ T cells (Fig. 5b right). Moreover, the contribution of infected CD20$^{dim}$ T cells to the total pool of HIV-infected cells significantly increased over time, reaching 30% at day 5 post-infection (Fig. 5c left). Of note, we also observed that CD20 expression was increased in the general population of T cells after infection (Fig. 5c right), indicating that

upregulation of CD20 in infected cells impacted the overall expression of CD20 in the cell culture. Notably, a significant positive correlation was found between the percentage of productive HIV-infected cells and the percentage of CD20$^+$ infected cells (Fig. 5d left), or with the mean fluorescence intensity of CD20 expression of the infected fraction (Fig. 5d right), suggesting a selective upregulation of CD20 in HIV-infected cells. In addition, in order to show the unequivocal upregulation of CD20 after HIV infection, we performed cell sorting of the CD4$^+$ CD20$^-$ population before cell infection. Again, a clear induction of CD20 expression after 3 days of cell infection was observed (Supplementary Fig. 3a, b). This result was confirmed by quantification of CD20-mRNA before and after cell infection using qPCR (Supplementary Fig. 3c). Since we have recently described that CD32 is also upregulated in HIV-infected cells[17], we next investigated the

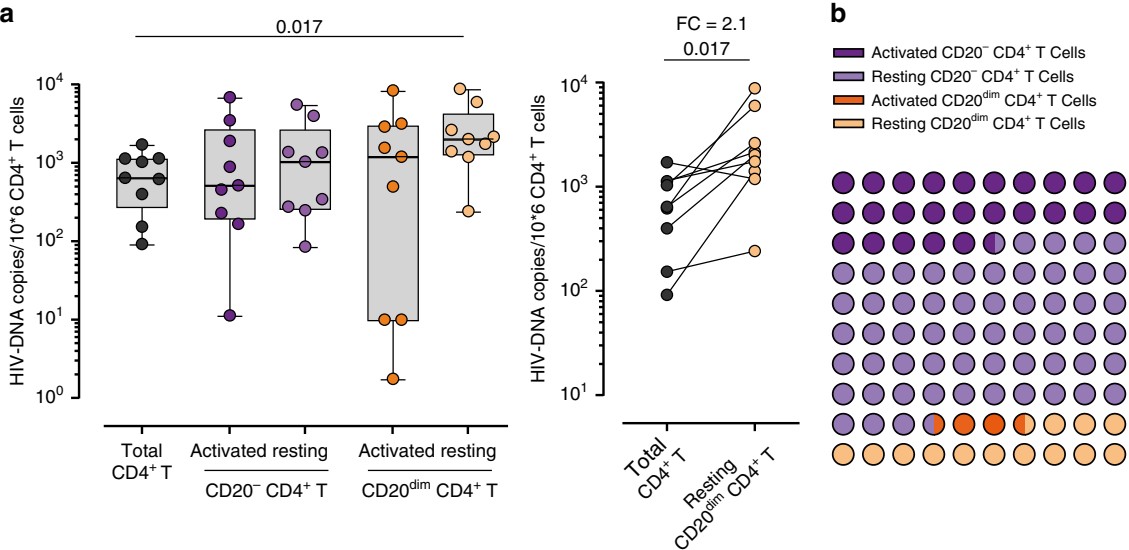

**Fig. 3** HIV-1 DNA quantification in CD20$^{dim}$ CD4$^+$ T cells. HIV DNA was measured in sorted CD4$^+$ T cells (black), activated CD20$^-$ CD4$^+$ T cells (dark violet), resting CD20$^-$ CD4$^+$ T cells (light violet), activated CD20$^{dim}$ CD4$^+$ T cells (dark orange), and resting CD20$^{dim}$ CD4$^+$ T cells (light orange) in PBMCs from nine ART-suppressed patients. The activation state was defined by the expression of HLA-DR, CD69 or CD25 markers. **a** Copies of viral DNA molecules per 10$^6$ CD4$^+$ T cells are represented for all sorted populations (left panel). Medians and min–max rank are represented. Comparison of HIV DNA copies per 10$^6$ cells between CD4$^+$ T cells and resting CD20$^{dim}$ CD4$^+$ T cells with the fold-change (FC, right panel). Comparisons were performed using the Wilcoxon test. **b** Contribution of each cell population to the total pool of HIV DNA. In panels (**a**) and (**b**), data from patients #11–14, 19, and 27–30 are shown. Data underlying this Figure are provided as Source Data file

relationship between these two markers. We observed that in HIV-infected cells, CD20 was upregulated to a greater extent than CD32 (Supplementary Fig. 4). Finally, we observed that CD20$^{dim}$ HIV-infected cells expressed more frequently HLA-DR than CD20$^-$ HIV-infected cells (ANOVA $p = 0.001$), and also the immune checkpoint marker PD-1 (ANOVA $p = 0.01$) (Fig. 5e). To ascertain the dynamics of CD20 expression in HIV-infected cells, we took advantage of the CD4 downregulation induced by HIV in highly productive HIV-infected cells. During early infection, defined by infected cells expressing CD4, levels of CD20 were notably higher than during late infection, as defined by cells with marked downregulation of the CD4 receptor (Fig. 5f). The opposite trend was observed for the activation marker HLA-DR (Supplementary Fig. 5a). These results suggest that this early upregulation of CD20 upon HIV infection might be a virus-specific effect, since the general activation of T cells with anti-CD3 and CD28 antibodies only increased the expression of CD20 over time (96 h) slightly. In contrast, CD69 and HLA-DR were strongly upregulated after CD3/CD28 activation (Supplementary Fig. 5b).

**Rituximab targets HIV-infected CD20$^{dim}$ CD4$^+$ T cells in vitro.** Rituximab is a chimeric monoclonal antibody directed against the B-cell-specific antigen CD20 used to deplete B cells in certain types of lymphoma[18]. We tested if Rituximab could also target and deplete T cells expressing low levels of CD20. PBMCs from uninfected donors and three ART-suppressed individuals (median time on suppressive ART of 336 months) (Supplementary Table 1) were treated with 10 μg/ml of Rituximab for 48 h. As expected, Rituximab significantly reduced the number of B cells (Fig. 6a). Moreover, a marked decreased of CD20$^{dim}$ CD4$^+$ T cells in culture was also observed (Fig. 6b). We also studied if Rituximab induced antibody-dependent cell-mediated cytotoxicity (ADCC) of CD20$^{dim}$ CD4$^+$ T cells. To achieve this goal, Rituximab-treated CD4$^+$ T cells, stained with the eFluor670 dye as previously shown[19], were used as target cells in combination with Natural killer (NK) effector cells. We found that NK cells were responsible for inducing a potent ADCC response that

significantly reduced the proportion of CD20$^{dim}$ CD4$^+$ T cells (Fig. 6c). Gating strategy and the individual data are shown in Supplementary Fig. 6.

Since we have observed that CD20 was upregulated upon HIV infection and Rituximab efficiently targeted CD20$^{dim}$ CD4$^+$ T cells, we next asked if Rituximab had any impact on ex vivo infection cultures of unstimulated PBMCs. Addition of Rituximab to the cell culture completely abrogated productive HIV infection of CD20$^{dim}$ CD4$^+$ T cells (Fig. 6d, left panel). It is noteworthy that the entire population of CD20$^{dim}$ CD4$^+$ T cells was not depleted by Rituximab (as observed in Fig. 6b); however, cells that were efficiently eliminated were those supporting productive HIV infection. Moreover, Rituximab impacted the total pool of infected CD4$^+$ T cells, reducing HIV infection by 38.6% at day 3 (Fig. 6d, right panel).

**In vivo administration of Rituximab targets HIV-RNA$^+$ cells.** To assess the in vivo effect of Rituximab on the HIV-reservoir, we took advantage of available samples obtained from three ART-suppressed patients receiving Rituximab for non-malignant reasons. Patient #52 received Rituximab at 1000 mg/dose on days 0 and 14 due to an IgG4-related systemic disease. Patient #53 only received one dose of 1000 mg due to fibrillar glomerulonephritis. Patient #59 received three doses of 375 mg/m$^2$ of Rituximab at days 0, 7, and 33 due to a mucocutaneous ulcer caused by Epstein–Bar virus. Consistent with our in vitro data, we observed that Rituximab efficiently depleted more than 80% of CD20$^{dim}$ CD4$^+$ T cells in the three patients, and this depletion was continually sustained throughout the study (Fig. 7a, b). Moreover, we measured intracellular HIV RNA in isolated CD4$^+$ T cells in these patients. Although no LRAs were administered to the patients, we observed a transient 2.38 fold-decrease of HIV RNA in one patient (#52), a sustained 2.4-fold reduction of HIV RNA in the second patient (#53), and no significant changes in patient #59 (Fig. 7c). The low basal HIV transcription level observed in the third patient might explain the lack of rituximab effect on HIV$^+$ cells. As body distribution of different cell subpopulations

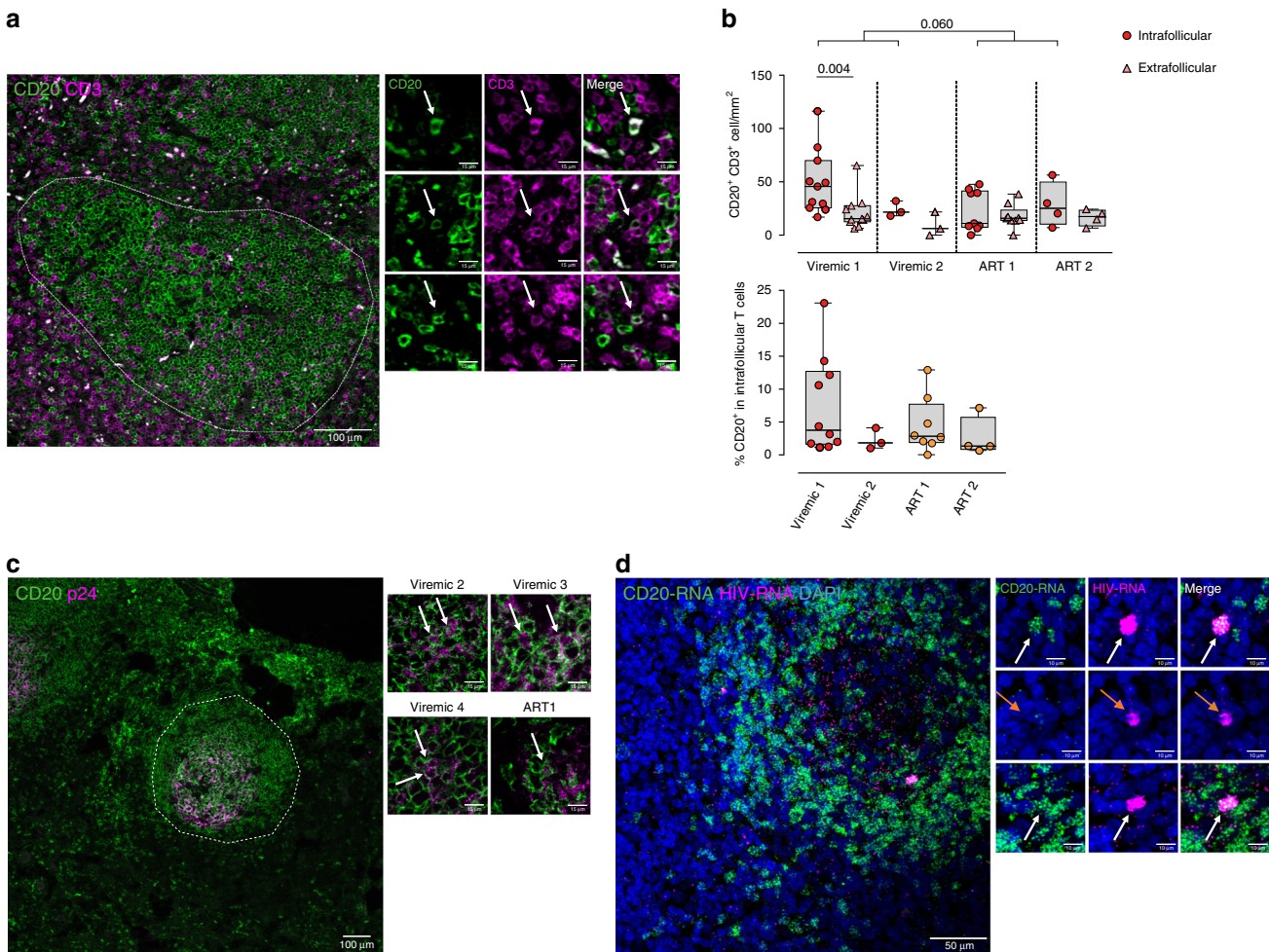

**Fig. 4** CD20 expression in T cells of intact human lymph nodes. Paraffin-embedded lymph node sections from HIV-infected patients were stained with anti-CD20 combined with anti-CD3 (**a**) or anti-p24 antibodies (**b**), alternatively sections were proceed for in situ RNA-hybridization of CD20-mRNA or mRNA and HIV RNA (**c**). **a** Representative micrograph of a lymph node section from one viremic HIV$^+$ patient stained with anti-CD20 (green) and anti-CD3 (violet) antibodies. The discontinuous line indicates a B-cell follicle and the white arrows double positive cells. Right panels correspond to zoomed views from different patients. **b** Upper panel shows the number of cells per mm$^2$ expressing CD20$^+$ and CD3$^+$ in lymph nodes (circles, each dot corresponds to one follicle) and in the extrafollicular region (triangles, each dot corresponds to one optical field). Two HIV-viremic (#84 and 85, in red) and two ART-suppressed patients (#31 and 32, in orange) are shown. Comparison between values inside B-cell follicles from viremic and ART-suppressed patients was performed using a Mann–Whitney test, and comparison within viremic 1 was performed with the Wilcoxon test. Lower panel shows the percentage of CD3$^+$ cells within the B-cell follicle expressing CD20. Each dot corresponds to one follicle. Medians and min–max rank are represented. **c** Representative micrograph from a lymph node section from one HIV-viremic patient stained with anti-CD20 (green) and anti-p24 (violet) antibodies. High magnification images correspond to B-cell regions from different ART-suppressed and viremic patients, white arrows indicate double positive cells and the discontinuous line indicates B-cell follicle. Patients #31 and 82–84 are shown. **d** Representative image of a B-cell follicle section from one viremic patient (#82) subjected to the in situ hybridization technique. CD20-mRNA is represented in green and HIV RNA in violet. At the middle, zoomed images of double positive cells (white arrows) or HIV RNA single positive cells (orange arrows). Data underlying this Figure are provided as Source Data file

might impact detection of HIV-RNA$^+$ cells in blood, we quantified the proportion of major CD4$^+$ T-cell subsets after Rituximab administration. No consistent changes in cell proportions were observed (Supplementary Fig. 7).

**Rituximab impacts HIV$^+$ cells after viral reactivation.** As the frequency of CD20 expression in CD4$^+$ T cells is relatively low, and HIV infection upregulates CD20, we wondered if reactivation of the latent reservoir might also upregulate CD20 in HIV-infected cells and render cells susceptible to Rituximab. Samples from ART-suppressed patients were included in these studies (median time on suppressive ART of 23 months) (Supplementary Table 1). First, we performed in vitro reactivation assays using fresh CD4$^+$ T cells from eight ART-suppressed HIV-infected patients. The LRAs romidepsin, ingenol or the positive control

PMA/ionomycin were added to cells and the expression of HIV RNA was assessed by the RNA FISH-flow assay (Fig. 8a). The addition of romidepsin and ingenol significantly increased the proportion of CD20$^{dim}$ cells expressing viral transcripts, with higher frequencies observed in CD20$^{dim}$ cells compared with CD20-negative cells in all conditions (Fig. 8b), and with a concomitant increase in the proportion of infected cells expressing CD20$^{dim}$ in the whole population of CD4$^+$ T cells (Fig. 8c). Under all conditions tested, cells expressing HIV RNA presented higher levels of CD20$^{dim}$ compared with uninfected cells. In addition, ingenol and PMA/ionomycin induced a significant upregulation of CD20$^{dim}$ in the whole population of infected CD4$^+$ T cells after viral reactivation. A very low upregulation of CD20 in uninfected cells and CD8 T cells was also observed for ingenol and PMA/ionomycin (Fig. 8d).

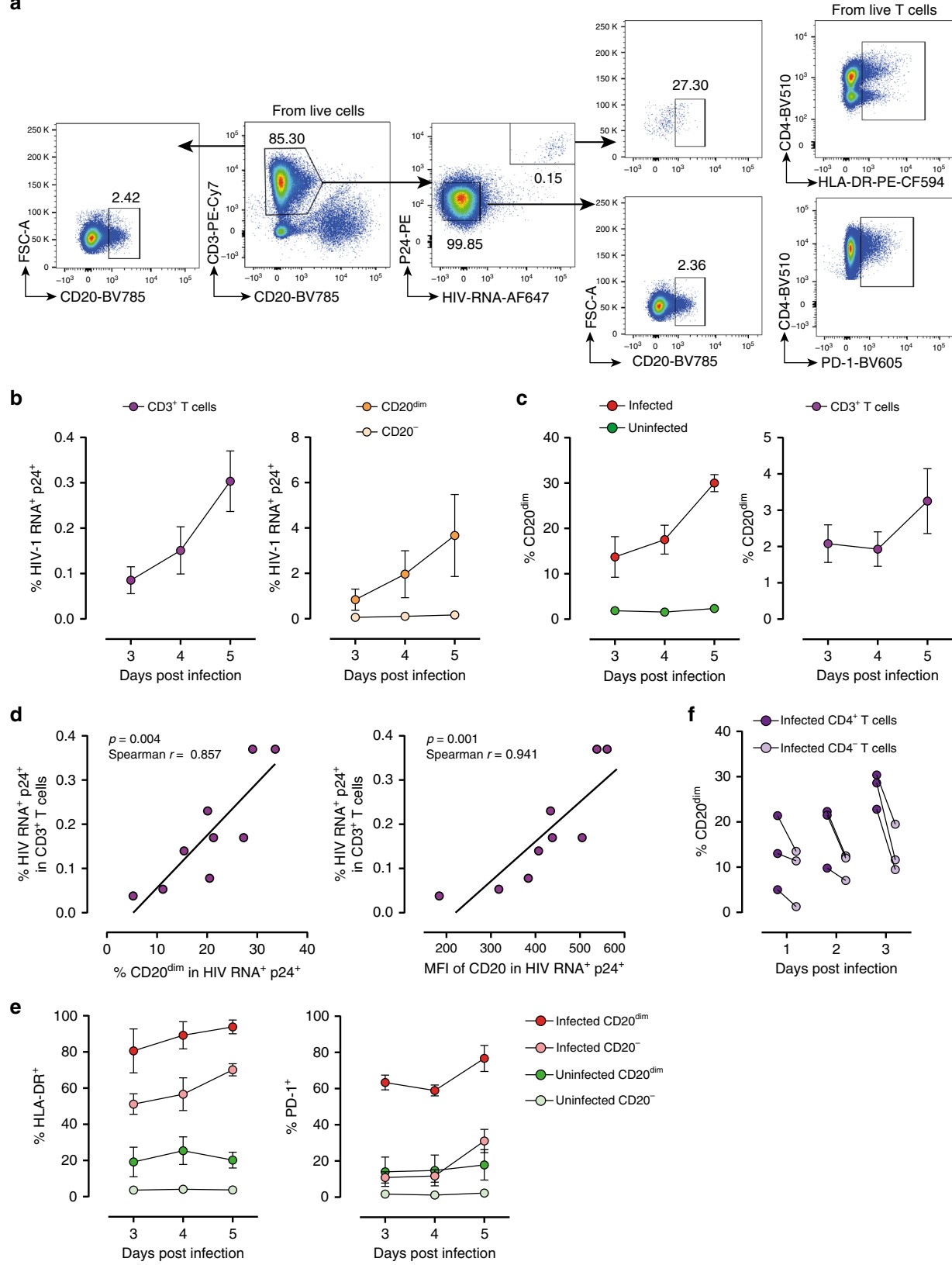

Next, we investigated whether Rituximab was able to impact cells actively transcribing HIV RNA after pharmacological viral reactivation in 12 ART-suppressed patients. PBMCs from these patients were incubated for 24 h with Rituximab and romidepsin or panobinostat, and for an additional 24 h with Rituximab.

Thereafter, CD4+ T cells were isolated and HIV RNA quantified by conventional qPCR. We observed successful viral reactivation in 7 out of 12 samples tested (median fold-change reactivation of 3.7). Importantly, the potency to induce viral reactivation by LRAs was directly associated with the fold-change reduction of

**Fig. 5** Ex vivo infection upregulates CD20 expression. Unstimulated PBMCs from three uninfected donors were infected with HIV strain NL4.3. Infection was monitored by simultaneous staining of HIV RNA using the RNA FISH-flow assay and the viral protein p24 at days 3, 4, and 5. **a** Gating strategy used to monitor HIV infection and expression of CD20$^{dim}$. Previous sequential gates are represented in Supplementary Fig. 9. **b** Percentage of productively HIV-infected cells in CD3$^+$ T cells (left panel) and in CD20$^{dim}$ or CD20$^-$ CD3$^+$ T cells (right panel). **c** Expression of CD20$^{dim}$ in infected and uninfected CD3$^+$ T cells (left panel) and in the total CD3$^+$ T cell population (right panel). **d** Correlation between the proportion of infected cells within the CD3$^+$ T cell population and CD20 expression (left panel) and the mean fluorescence intensity (MFI) of CD20 (right panel). **e** Percentage of HLA-DR and PD-1 in infected and uninfected cells expressing or not expressing CD20$^{dim}$. **f** Proportion of CD20$^{dim}$ in infected cells expressing the CD4 cell receptor versus infected cells with marked downregulation of the CD4 receptor. In all panels, the mean and SEM value of three independent experiments is represented. In panel (**d**), Spearman's nonparametric correlation coefficients and associated $p$ values are shown. Data underlying this Figure are provided as Source Data file

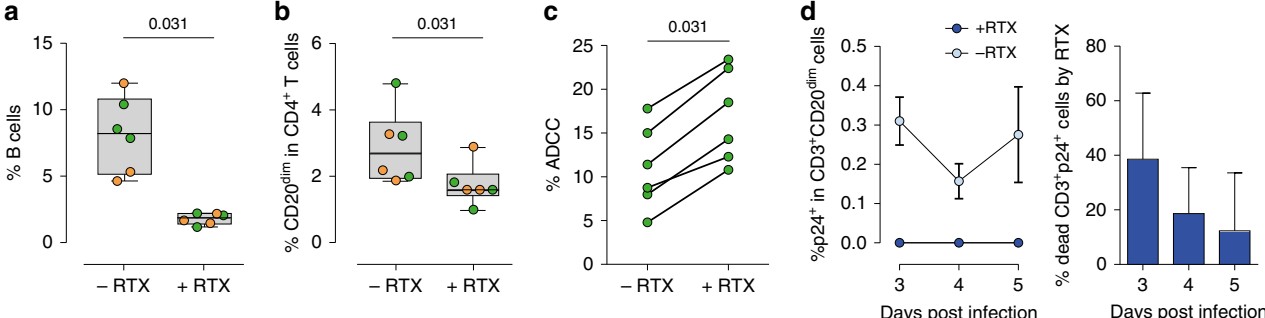

**Fig. 6** Rituximab treatment eliminates infected CD20$^{dim}$ CD4$^+$ T cells and reduces HIV-1 infection in vitro. **a, b** Isolated PBMCs from uninfected donors (green) and ART-suppressed HIV$^+$ patients (orange) were treated with Rituximab (RTX) or antibody control for 48 h. Cell death was assessed by flow cytometry using the gating strategy shown in Fig. 1a. Panels represent percentages of B cells (**a**) or CD20$^{dim}$ in CD4$^+$ T cells (**b**) for $n = 3$ uninfected donors and $n = 3$ ART-suppressed patients (#56–58). Median values and min–max ranks are represented. **c** Isolated CD4$^+$ T cells incubated with Rituximab or antibody control were used as a target in combination with isolated autologous NK cells. The percentage of ADCC was measured by the reduction of eFluor670 staining by flow cytometry using the gating strategy shown in Fig. S6. Summary results for six uninfected donors are represented. **d** PBMCs from $n = 3$ uninfected donors were infected with the HIV strain NL4.3 and treated with Rituximab for different periods of time. Percentages of infected cells (p24$^+$) in CD20$^{dim}$ CD3$^+$ T cells with (dark blue) or without (light blue) Rituximab are represented in the left panel (mean ± SEM). The percentage of dead infected cells induced by Rituximab is shown in the right panel, which represents the difference in the proportion of infected cells between Rituximab-treated cells versus Rituximab non-treated cells (mean ± SD). Flow gating strategy used for this analysis is shown in Fig. 5a. All comparisons were performed using the Wilcoxon test. Data underlying this Figure are provided as Source Data file

RNA-expressing cells by Rituximab (Fig. 8e). Moreover, in the seven samples responding to LRA viral reactivation, Rituximab induced significant depletion of HIV RNA-expressing cells compared with cells non-responding to LRAs (Fig. 8f), with a median reduction of HIV RNA of 64% [18–87]. For samples responding to LRA treatment, individual patterns of viral RNA expression in the presence of LRAs with or without Rituximab are shown in Supplementary Fig. 8. Overall, Rituximab was capable of targeting and deplete cells expressing HIV RNA after viral reactivation. However, the ability of Rituximab to impact this transcriptionally active viral reservoir directly depended on the ability of LRAs to induce viral transcription.

## Discussion

Understanding the mechanisms that contribute to HIV-1 persistence despite ART, and the development of therapeutic strategies to target persistent virus, represent current priorities in HIV-1 research[20,21]. Here we report the existence of a significant fraction of circulating CD4$^+$ T cells that express CD20, a cell membrane receptor preferentially expressed on B cells, which associates with HIV infection and cells enriched for viral RNA in ART-suppressed patients. Importantly, treatment with Rituximab eliminates this fraction while impacting on the HIV-viral reservoir when combined with LRAs.

Using different methodologies we unequivocally demonstrated that CD20$^{dim}$ CD4$^+$ T cells are a truly single cell fraction, and they are not the result of flow cytometry artifacts, as previously

shown for T cells expressing high levels of CD32[22]. We found that a small proportion of CD4$^+$ T cells expressed dim levels of CD20, and this population was significantly expanded by ~2-fold in viremic patients. These findings, together with the enrichment of HIV RNA in this subpopulation, the upregulation of CD20 after in vitro infection, and the normalization of CD20 levels in ART-suppressed individuals, strongly suggests a direct link between HIV infection and the expression of CD20. Moreover, the frequency of CD4$^+$ T cells expressing CD20 inversely correlated with CD4$^+$ T-cell counts and the time on suppressive ART. This phenomenon has been extensively reported for conventional activation markers, such as CD38 and HLA-DR[23,24]. Thus, it is tempting to speculate that CD20 might behave as a non-conventional activation marker during HIV infection for CD4$^+$ T cells. We found that in general, ~1% of CD4$^+$ T cells expressed CD20, and most of those cells had a memory phenotype. These percentages of expression have been previously reported in patients with multiple sclerosis and rheumatoid arthritis[9,25]. In agreement with our results, Schuh and colleagues studied features of human CD3$^+$ CD20$^{dim}$ cells during multiple sclerosis and found that these T cells were enriched in CD45RO$^+$ memory cells[25]. We also found that CD4$^+$ T cells expressing the CD20 marker were in general more activated than their CD20$^-$ counterparts. Accordingly, increased activation of T cells expressing CD20 has been widely reported[26].

The identification of specific cell subpopulations harboring HIV is important for the design of novel therapies directed to

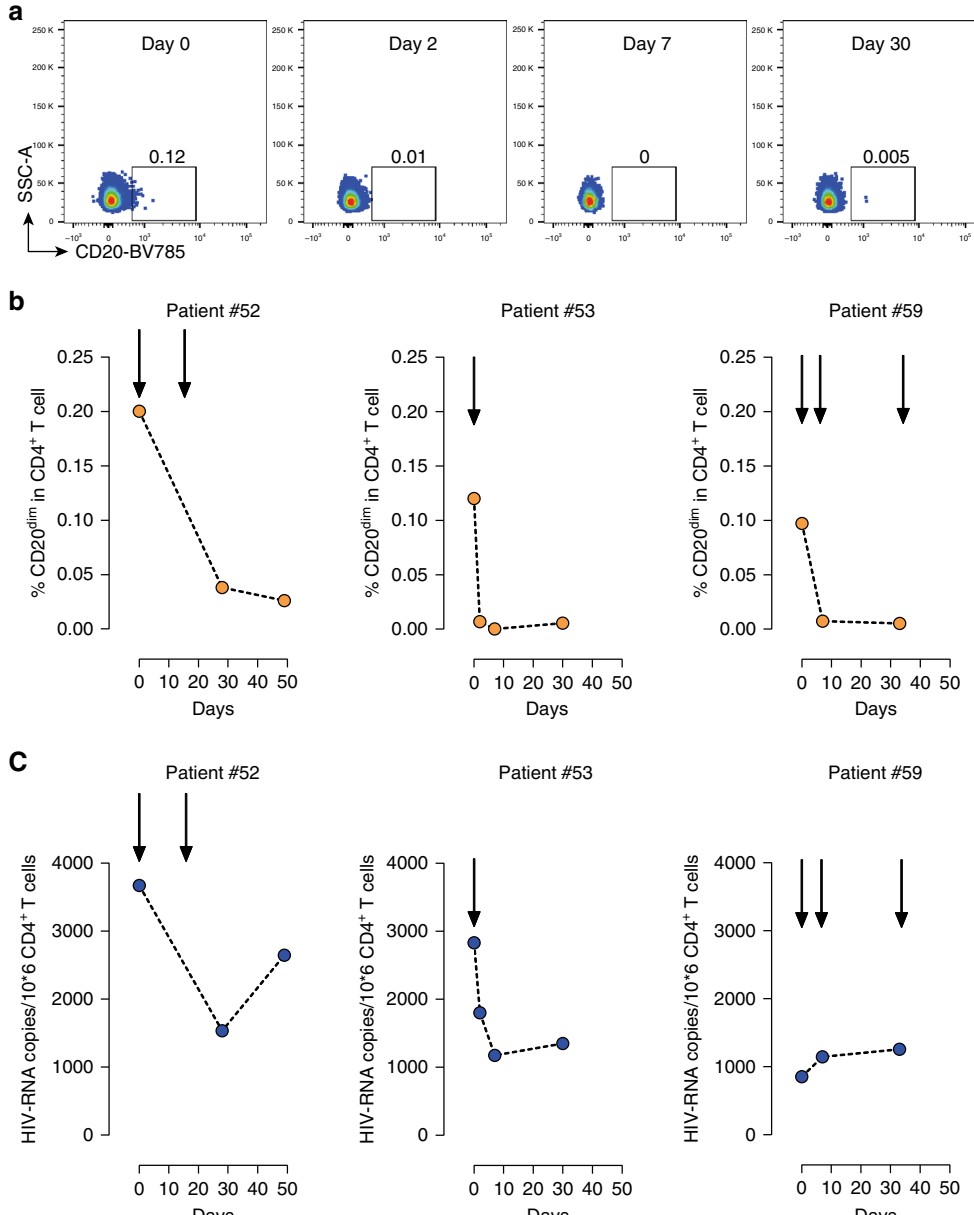

**Fig. 7** Effects of Rituximab treatment in CD20$^{dim}$ CD4$^+$ T cells and intracellular HIV RNA. Three ART-suppressed patients were treated with Rituximab at indicated times (arrows) and blood samples were longitudinally collected. **a** Flow plots, following the same gating strategy as shown in Fig. 1a, of CD20 expression in CD4$^+$ T cells in patient #53 are shown. **b** Frequencies of CD20$^{dim}$ CD4$^+$ T cells in longitudinal samples from three patients are shown. **c** Copies of HIV RNA per million CD4$^+$ T cells after Rituximab treatment. Individuals #52, #53, #59 are represented. Data underlying this Figure are provided as Source Data file

deplete the latent viral reservoir. We observed that CD20$^{dim}$ CD4$^+$ T cells were not particularly enriched in HIV DNA, compared with levels reported in other memory subsets[27,28]. However, we observed a significantly higher proportion of these cells expressing HIV RNA transcripts with a significant contribution to the total pool of HIV RNA-expressing cells. This result concur with HIV-RNA expression by other memory pools[4] and with the higher activation status of CD20$^{dim}$ cells. Other markers for transcriptionally active HIV-infected cells have been described recently: CD32[17,29,30] and CD30[31]. A common link between three markers, CD20, CD30 and CD32, is their relationship with increased levels of cell activation. However, we found that a minority of HIV-infected cells co-expressed CD20 and CD32 in blood. Thus, although CD20 and CD32 were both upregulated in HIV-infected cells, we speculate that the expression kinetics and/

or its cellular origin might be different. To date, the mechanism and function of these two markers in transcriptionally active HIV-infected cells remain unknown and merit further investigation.

In our in vitro model of HIV infection we showed that HIV replication increased the expression of CD20. Although only 30% of HIV-infected cells expressed CD20, a clear association was observed between infection and proportion of CD20. For instance, we observed that CD20 upregulation was stronger during the early events of HIV infection. Importantly, this result might explain the lack of CD20 expression in all HIV-infected cells analyzed. Based on these results, we conclude that CD20 expression is specially upregulated during HIV infection, but also it might slightly increase upon cell activation. We present several evidences supporting this argument: (i) we demonstrated that stimulation of

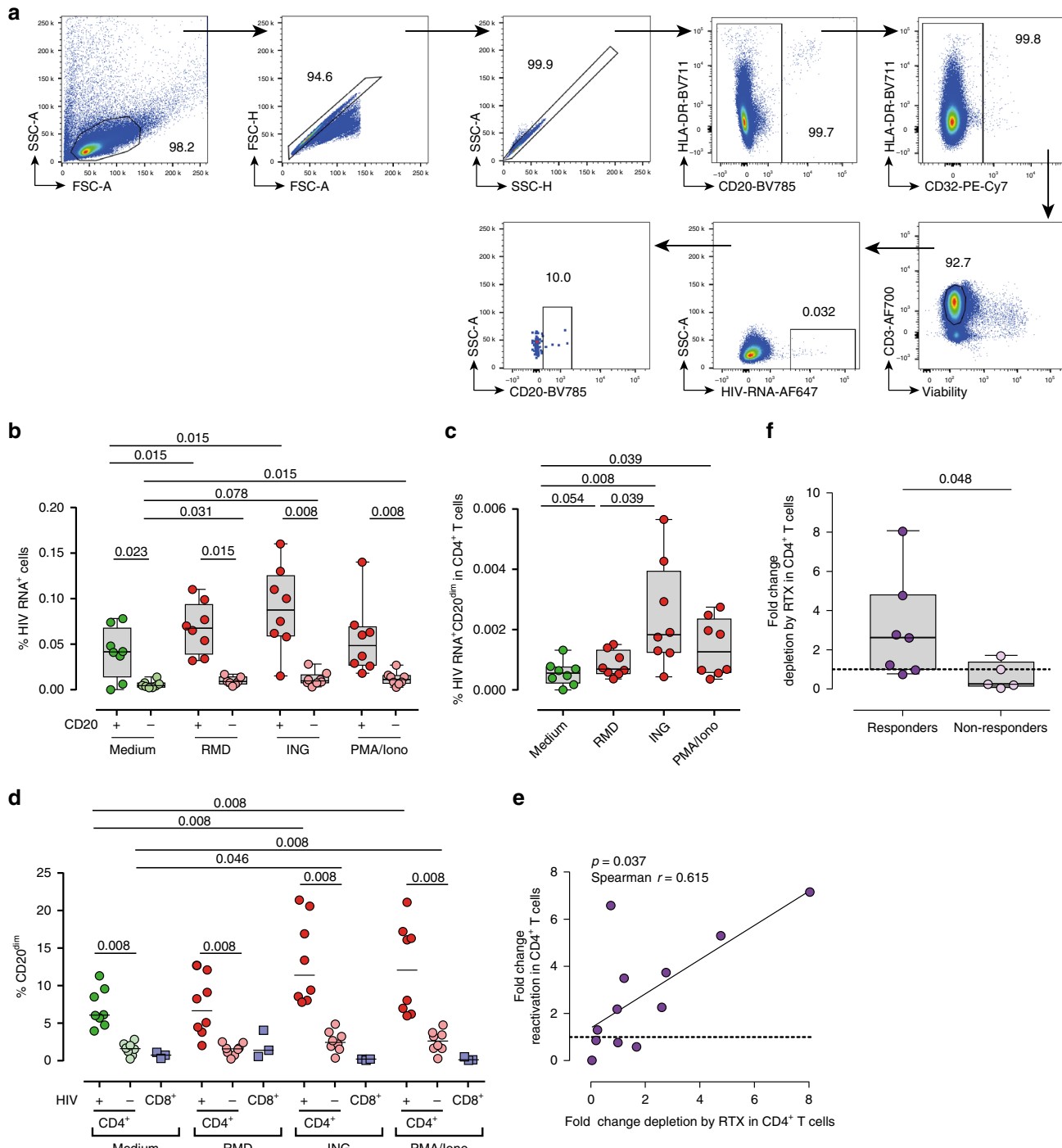

**Fig. 8** Rituximab impacts pharmacologically HIV-reactivated cells. **a–c** Freshly isolated CD4[+] cells from (#33–40) and CD8[+] cells (#17–19) from $n = 8$ and $n = 3$ ART-suppressed patients, respectively, were treated with romidepsin (RMD), ingenol (ING), or PMA/ionomycin (PMA/Iono) for 22 h and subjected to the RNA FISH-flow assay for the detection of HIV RNA transcripts. **a** Representative flow gating strategy used for the analysis. Proportions of cells expressing HIV RNA within CD20[dim] and CD20-negative CD4[+] T cells (**b**) or cells expressing HIV RNA and CD20[dim] in CD4[+] T cells (**c**) are shown. Medians and min–max rank are represented. The median percentages of CD20[dim] in infected and uninfected CD4[+] T cells, and CD8[+] T cells after treatment with different LRAs are shown in (**d**). Comparisons were performed using the Wilcoxon test. **e**, **f** Unfractionated PBMCs from $n = 12$ ART-suppressed patients (#18, 19, 21, 23, 41–48) were treated with the latent reversal agents (LRAs) romidepsin or panobinostat for 24 h, and with Rituximab for 48 h. HIV RNA expression were measured by qPCR. Panel (**e**) shows the correlation between the fold-change reactivation by the LRA and the fold-change cell depletion by Rituximab. Spearman's nonparametric correlation coefficient and associated $p$ value are shown. Panel (**f**) shows the fold-change in cell depletion by Rituximab in patients, stratified by responders (fold-change reactivation by LRA > 1) and non-responders (fold-change reactivation by LRA < 1). The Mann–Whitney test was used for statistical analyses in all panels. Median and min–max ranks are represented in all panels. Data underlying this Figure are provided as Source Data file

uninfected donors with anti-CD3 and anti-CD28 antibodies slightly promotes CD20 expression in uninfected CD4[+] T cells (up to 4%) (Supplementary Fig. 5b); (ii) LRA treatment slightly increases the expression of CD20 in HIV-negative cells (up to 5%), but the upregulation is stronger in HIV[+] cells (up to 20%) (Fig. 8d); and (iii) ex vivo HIV infection of FACS-sorted CD4[+]CD20[−] negative cells strongly upregulates CD20 (~30%) (Supplementary Fig. 3). Thus, it is tempting to speculate that viral proteins, i.e., Nef, which has been shown to modulate T-cell activation[32], may be involved in this process. More mechanistic studies are needed to finally elucidate the real contribution of the virus to CD20 upregulation. In concordance with this idea, we found higher expression of CD20 in HIV-infected cells in lymph nodes, a key sanctuary viral reservoir[33–36]. Thus, it is tempting to speculate that the infected cells expressing CD20 that we observed in lymph nodes are T$_{FH}$, a subpopulation of cells found within B-cell follicles with elevated expression levels of PD-1 and CXCR5, which are highly permissive to HIV infection and significant contributors to HIV persistence during treatment[37]. In agreement with that, we observed higher proportions of CD20[dim] CD4[+] T cells expressing PD-1 and CXCR5 in blood compared with cells lacking CD20 expression, and follicular T cells expressing CD20 in lymph nodes. This may suggest that the origin of this population may be the lymph nodes. If this hypothesis is fully confirmed in additional studies, targeting this subpopulation of HIV-infected cells will be fundamental to impact HIV persistence.

Critically, we found that Rituximab was able to target CD4[+] T cells expressing dim levels of CD20 and induced ADCC mediated by NK cells. A number of studies have also demonstrated the effects of anti-CD20 therapy on the T-cell compartment[9,38–40]. Importantly, using our in vitro HIV infection method, we observed that HIV-infected cells expressing CD20 were successfully eliminated by Rituximab. This finding suggests that Rituximab treatment during HIV infection might have a positive effect on controlling viral load. In this regard, in untreated SIV-infected rhesus monkeys, Rituximab administration prior to and during SIV infection was associated with decreased viral load. Interestingly, during chronic follow-up infection, the viral load was undetectable in six out of seven animals treated with Rituximab[41]. However, in another study with African green monkeys, depletion of CD20[+] T cells had no significant impact on viral load[42]. In humans, a case report described the impact of Rituximab alone on HIV viremia; HIV plasma viral load was maintained while on ART and Rituximab treatment. After therapy discontinuation, however, the HIV-viral load was significantly increased, most likely due to a loss of antibody production[43].

The use of anti-CD20 therapy to reduce viral reservoirs in B-cell follicles had been previously considered[44]; the destruction of B-cell follicles using anti-CD20 monoclonal antibodies has been proposed as a novel strategy to expose HIV-1 reservoirs for elimination. For instance, in a recent study, HIV-infected patients with lymphoma or Kaposi Sarcoma received multiple doses of Rituximab, but reductions in cell-associated HIV RNA or HIV DNA levels were not observed[45]. Of note, these patients were treated with a combination of different antineoplastic drugs, which included Rituximab. In our study, in two patients treated only with Rituximab for non-malignant reasons we observed more than twofold reductions in HIV RNA. Note that, in spite of cellular associated HIV RNA might naturally fluctuate during time in ART-treated patients, only 5% of patients are likely to present a random twofold variation in HIV-RNA levels[46]. Ongoing viral replication in particular tissues, i.e., lymph nodes, potentially impacted by Rituximab might help to explain the reduction in HIV RNA. Patient #53 was under a dual antiretroviral therapy, instead of the recommended triple therapy

regimen, which might help to explain the sustained reduction of HIV RNA observed in this patient.

Furthermore, our ex vivo data indicate that viral reactivation of latent cell reservoirs increased the levels of cells co-expressing HIV RNA and CD20, and consequently, the addition of Rituximab induced depletion of viral-reactivated cells. Based on this data, we speculate that a large in vivo reactivation of the latent viral reservoir, i.e., using LRAs, may be an indispensable prerequisite for Rituximab to significantly impact long-term HIV persistence in vivo. Although it is important to consider that a possible complication of current anti-CD20 therapy could be immunodeficiency and/or immunopathologies, we provide evidence for the combined use of LRAs and anti-CD20 therapies for the depletion of viral-reactivated cells in patients on ART. The future development of new anti-CD20 therapies with less toxicity and specifically directed to CD4[+] T cells might help to implement this new therapeutic approach.

## Methods

**Ethics statement**. PBMCs from HIV-1-infected patients were obtained from the HIV unit of the Hospital Universitari Vall d'Hebron and from the Clinic Hospital in Barcelona, Spain. Study protocols were approved by the Comitè d'Ètica d'Investigació Clínica (Institutional Review Board numbers 39–2016, 196–2015, and 476–2018) of the Hospital Universitari Vall d'Hebron and the Hospital Clinic in Barcelona, Spain. Samples were obtained from adults, all of whom provided written informed consent, and were prospectively collected and cryopreserved in the Biobanc (register number C.0003590). All samples received were completely anonymous and untraceable.

**Study samples**. Samples from HIV-1-infected patients with CD4[+] T-cell counts higher than 100 cells/mm[3] recruited in the HIV unit of the Hospital Universitari Vall d'Hebron and the Clinic Hospital in Barcelona, Spain, were included in this study. Information on plasma viral loads, CD4[+] T-cell counts, and time on ART from suppressed patients is summarized in Supplementary Table 1.

**Cells and virus**. PBMCs were obtained from HIV-infected patients by Ficoll-Paque density gradient centrifugation and cryopreserved in liquid nitrogen. PBMCs from uninfected donors were obtained anonymously from the BST (Banc de Sang i Teixits, Barcelona, Spain) and isolated as described above. PBMCs were cultured in RPMI medium (Gibco) supplemented with 10% Fetal Bovine Serum (Gibco), 100 μg/ml streptomycin (Fisher Scientific), and 100 U/ml penicillin (Fisher Scientific) (R10 medium).

The plasmid encoding HIV-1 strain NL4.3 (pNL4.3) was obtained through the NIH AIDS Reagent Program from Malcom Martin. Viral stocks were generated by transfection of 293T cells (ATCC, CRL-3216) with pNL4.3, and the resulting viral particles were titrated in TZMbl cells (NIH AIDS Reagent Program) using an enzyme luminescence assay (britelite plus kit; PerkinElmer) as described previously[47].

**Phenotyping of CD4[+] T cells**. To identify CD4[+] T-cell subsets and the expression of CD20[dim] cells, PBMCs were stained for cell surface markers in a final volume of 100 μl with CD3 (PE-Cy7, 2 μl, BD 557851), CD4 (AF700, 1.25 μl, BD 557922), CD20 (BV785, 2.5 μl, Biolegend 302356), HLA-DR (PE, 2.5 μl, Biolegend 63-9956-41), CD25 (PE, 1 μl, Biolegend 356103), CD69 (PE, 2.5 μl, Biolegend 310905), CD27 (FITC, 4 μl, BD 555440), CD45RO (BV605, 1.2 μl, Biolegend 304238), CCR7 (BV711, 1 μl, BD 566602 or PE-CF594, 1 μl, BD 150503) and CD95 (PE-Cy5, 10 μl, BD 559773) antibodies. Different CD4[+] T-cell subset phenotypes were identified as follows: Naive (NA): CD3[+], CD4[+], CCR7[+], CD45RO[−], CD27[+]; stem cell memory (SCM): CD3[+], CD4[+], CCR7[+], CD45RO[−], CD27[+], CD95[+]; central memory (CM): CD3[+], CD4[+], CCR7[+], CD45RO[+]; transitional memory (TM): CD3[+], CD4[+], CCR7[−], CD45RO[+], CD27[+]; effector memory (EM): CD3[+], CD4[+], CCR7[−], CD45RO[+], CD27[-], and terminal differentiated (TD): CD3[+], CD4[+], CCR7[−], CD45RO[−]. To identify CXCR5 and PD-1 expression in CD20[dim] T cells, PBMCs were surface stained in 100 μl with CD3 (PE-Cy5, 0.6 μl, Biolegend 300410), CD4 (AF700, 1.25 μl, BD 557922), CD20 (BV785, 2.5 μl, Biolegend 302356), CD19 (FITC, 2.5 μl, BD 560994), CXCR5 (PE, 2 μl, Biolegend 356903), and PD-1 (PE-Cy7, 0.6 μl, BD 561272) antibodies. Cell viability was determined with a violet viability dye for flow cytometry (LIVE/DEAD Fixable Violet Dead Cell Stain Kit; Invitrogen L34963). All samples were acquired with a LSRFortessa flow cytometer (Becton Dickinson), and the results were analyzed with FlowJo v10 software (Tree Star).

For fluorescent flow cytometry imaging analyses, PBMCs were stained in 100 μl with CD3 (PE-Cy7, 2 μl, BD 557851), CD4 (APC, 5 μl, ThermoFisher 17-0048-42), CD20 (FITC, 2 μl, Biolegend 302303), CD19 (PE, 0.6 μl, Biolegend 302207), and viability dye (LIVE/DEAD Fixable Violet Dead Cell Stain Kit, Invitrogen L34963).

Cells were acquired with an AMNIS ImageStremX imaging flow cytometer (Merck), and data was analyzed using IDEAS v6.1 software. Only focused cells were considered for analysis, gradient RMS value > 40 was established as a threshold for best focus.

**Quantification of HIV DNA and HIV RNA by quantitative PCR**. In accordance with the manufacturer's instructions, CD4$^+$ T lymphocytes were enriched from total PBMCs by negative selection (MagniSort Human CD4$^+$ T Cell Enrichment, eBioscience) when appropiate. CD4$^+$ T cells were subjected to RNA extraction using the mirVana™ miRNA Isolation Kit following the protocol instructions (Ambion). Reverse transcription of RNA to cDNA was performed with SuperScriptIII Reverse Transcriptase (Invitrogen) with random primers, and cDNA was quantified by qPCR with the following primers and probe: LTR R: 5′ GTTCGGGCGCCACTGCTAG 3′; LTR F: 5′ TTAAGCCTCAATAAAGCTTGCC 3′ and LTR Probe: 5′ /56-FAM/ CCAGAGTCA/ZEN/CACAACAGACGGGCA/31ABkFQ/ 3′. Quantification of RNA copies was performed using a standard curve, and values were normalized to 1 million CD4$^+$ T cells. In addition, CD4$^+$ T cells were used to quantify total HIV DNA with the same primers and probe detailed above. CD4$^+$ T cells were immediately lysed with proteinase K-containing lysis buffer, and HIV DNA was quantified. Contribution of each subset to the total HIV-reservoir was calculated by considering the frequency of each subset (analyzed by flow cytometry) within the total CD4 compartment and their relative infection frequency.

**Detection of HIV RNA$^+$ cells by the RNA FISH-flow assay**. The RNA FISH-flow assay was performed according to manufacturer's instructions (Human Primer-Flow RNA Assay, eBioscience) with some modifications. After antibody staining and cell fixation and permeabilization, cells were ready for hybridization with a set of 50 probes spanning the whole *Gag-Pol* HIV mRNA sequence (bases 1165 to 4402 of the HXB2 consensus genome). Next, the cells were subjected to amplification signal steps, and HIV RNA was detected using Alexa Fluor 647-labeled probes. As a negative control, non-HIV-infected donors were included in each experiment; to normalize the data, the percentage of HIV RNA$^+$ cells detected in donor samples was subtracted to the real values obtained in the HIV-infected samples. In these experiments, the following antibody panel was used in 100 μl: CD3 (PE-Cy5, 0.6 μl, BD 300410), CD4 (BV510, 1.25 μl, BD 344633), CD20 (BV785, 2.5 μl, Biolegend 302356), HLA-DR (PE, 2.5 μl, Biolegend 307605), CD25 (PE, 1 μl, Biolegend 356103), and CD69 (PE, 2.5 μl, Biolegend 310905). Cell viability was analyzed with a violet viability dye (Invitrogen, L34963). Samples were analyzed with the LSRFortessa flow cytometer, and the results analyzed with FlowJo v10 software.

**Fluorescence activated cell sorting of CD4$^+$ T cells**. For the sorting experiments, 100 million PBMCs surface stained in 100 μl with the following: CD3 (PerCP, 5 μl, BD 340663), CD4 (AF700, 1.25 μl, BD 557922), CD20 (FITC, 2 μl, Biolegend 302303), HLA-DR (PE, 2.5 μl, Biolegened 307605), CD69 (PE, 2.5 μl, Biolegend 310905), and CD25 (PE, 1 μl, Biolegend 356103) or CD3 (PE-Cy7, 2 μl, BD 557851), CD4 (AF700, 1.25 μl, BD 557922), CD20 (FITC, 2 μl, Biolegend 302303), and CD19 (PE, 0.6 μl, Biolegend 302207) antibodies, and both stained with a violet viability dye (Invitrogen, L34963). Cells were immediately sorted using a BD FACSAria Cell Sorter (Flow Cytometry Platform, Institut d'Investigació en Ciències de la Salut Germans Trias i Pujol, IGTP). For HIV DNA measurements, different sorted population were defined as follows: total CD4$^+$: CD3$^+$, CD4$^+$, CD20$^-$; CD4$^+$ activated: CD3$^+$, CD4$^+$, CD20$^-$, HLA-DR$^+$, CD69$^+$, CD25$^+$; CD4$^+$ resting: CD3$^+$, CD4$^+$, CD20$^-$, HLA-DR$^-$, CD69$^-$, CD25$^-$; CD20 activated: CD3$^+$, CD4$^+$, CD20$^{dim}$, HLA-DR$^+$, CD69$^+$, CD25$^+$; CD20 resting: CD3$^+$, CD4$^+$, CD20$^{dim}$, HLA-DR$^-$, CD69$^-$, CD25$^-$. After cell isolation, cells were subjected to HIV DNA quantification by qPCR as described above. For CD20-mRNA measurement, the following populations were isolated: CD4$^+$ CD20$^-$ T cells: CD3$^+$, CD4$^+$, CD20$^-$; CD4$^+$ CD20$^{dim}$ T cells: CD3$^+$, CD4$^+$, CD20$^{dim}$; B cells: CD3$^-$, CD4$^-$; CD20$^+$, CD19$^+$. Total RNA was extracted and cDNA was generated as described above. Levels of CD20-mRNA were relatively quantified using the β-actin reference gene (PerfeCTa® qPCR FastMix II®, ROX, Quantabio) and applying the $2^{-\Delta\Delta CT}$ method.

**Immunohistochemistry of human lymph nodes**. For immunodetection of CD20, CD3, and p24, formalin-fixed and paraffin-embedded (FFPE) sections from human lymph nodes of HIV-infected patients were dewaxed and placed in decreasing ethanol concentrations. Heat-induced epitope retrieval was performed by auto-claving at 120 °C for 15 min in a citrate buffer pH 6 (Abcam). Then, the slides were permeabilized in 1X Tris-buffered saline (TBS) (Fisher scientific) with 0.1% Triton X-100 (Sigma-Aldrich) and 1% BSA (Sigma-Aldrich) for 10 min. Subsequently, blocking was added for 2 h with 1X TBS supplemented with 10% normal donkey serum (Jackson Immunoresearch) and 1% BSA. The slides were incubated with primary antibodies overnight at 4 °C: anti-CD20 (goat-polyclonal anti-CD20 antibody, 1/100, Abcam ab194970), anti-CD3 (mouse-monoclonal anti-CD3 antibody, 1/100, Leica Biosystems NCL-L-CD3-565), or anti-p24 (mouse-monoclonal anti-p24 antibody, 1/10, Dako-Agilent M0857), diluted in TBS 1× −1% BSA. Next, samples were washed and incubated for 1 h with the appropriate

secondary antibody: Alexa Fluor 546 donkey anti-goat (Invitrogen, 712-586-150) and Alexa Fluor 647 donkey anti-mouse (Invitrogen, A-31571), counterstained with DAPI (4′,6-diamidino-2-phenylindole-dilactate, ThermoFisher), and mounted with Fluoromount G (eBioscience).

**Detection of RNA in lymph nodes by in situ hybridization**. We used the ultrasensitive ViewRNA ISH Tissue 2-Plex Assay Kit (eBioscience) to detect HIV RNA and CD20-mRNA in lymph node tissue samples. Briefly, tissue was fixed and embedded in paraffin. Sections were mounted on Superfrost Plus microscope slides (Fisher Scientifics). Before the assay was performed, samples were dewaxed in Xylene and dehydrated with ethanol. Sample pretreatment was performed by boiling sections in pretreatment solution for 10 min and protease digestion for 20 min at 40 °C. Hybridization was carried out by slide incubation with target probes for 2 h at 40 °C. Next day, signal amplification was performed by sequential incubation of Pre-Amplifiers, Amplifiers and label probes. Finally, samples were counterstained with DAPI.

**Confocal microscopy and quantification**. Samples were imaged on an Olympus Spectral Confocal Microscope FV1000 using a ×20 and ×40 phase objective and sequential mode to separately capture the fluorescence from the different fluor-ochromes at an image resolution of $800 \times 800$ pixels. ImageJ software was used to perform image processing and analysis. For CD20/CD3 colocalization analysis, a binary mask was generated for each channel, and the "analyze particles" tool was used to count single and double-stained cells. In all experiments, at least 10 B-cell follicles were analyzed per patient.

**Ex vivo infection of unstimulated PBMCs**. PBMCs from uninfected donors were thawed and incubated overnight in R10 with 40 U/ml interleukin-2 (IL-2). The next day, PBMCs were incubated for 4 h at 37 °C with 350,000 TCID$_{50}$ (50% tissue culture infectious dose) of NL4.3 viral strain. Cells were then thoroughly washed and cultured in 96-well plates with R10 containing 100 U/ml IL-2 for the next 6 days. To follow HIV RNA expression, on days 3, 4, and 5, at least 5 million cells were collected and subjected to the RNA FISH-flow assay. Target-specific Alexa Fluor 647-labeled probes were used to detect HIV-1-RNA; the expression of Gag p24 viral protein was determined using a p24-PE antibody (Beckman Coulter). Surface staining was performed in 100 μl with CD4 (BV510, 1.25 μl, Biolegend 344633), CD3 (PE-Cy5, 0.6 μl, Biolegend 300410), CD20 (BV785, 2.5 μl, Biolegend 302356), HLA-DR (Pe-Dazzle 594, 2.5 μl, BioLegend 307653), and PD-1 (BV605, 2.5 μl, BD 563245).

In the experiments with Rituximab treatment during ex vivo infection, at 4 h after infection cells were cultured with R10 supplemented with 100 U/ml IL-2 and 10 μg/ml Rituximab, or 10 μg/ml control antibody. At day 3, Rituximab (10 μg/ml) and the control antibody (10 μg/ml) were replaced. Infection was analyzed by p24-PE intracellular staining at days 3, 4, and 5. Surface staining was performed in 100 μl with CD3 (PE-Cy7, 2.5 μl, BD 557851), CD4 (AF700, 1.25 μl, BD 557922), CD20 (BV785, 2.5 μl, Biolegend 302356), and CD19 (FITC, 2.5 μl, BD 560994). All samples were acquired on a LSRFortessa and analyzed with FlowJo v10 software.

**Ex vivo infection of CD20$^-$ CD4$^+$ T cells**. Seventy million PBMCs were used for cell sorting experiments. Briefly, cells were stained in 100 μl with antibodies recognizing CD3 (PE-Cy7, 2.5 μl, BD 557851), CD4 (AF700, 1.25 μl, BD 557922), and CD20 (FITC, 2 μl, Biolegend 302303). A violet dye was used as a viability marker (Invitrogen, L34963). The population CD20$^-$ CD4$^+$ CD3$^+$ was immediately sorted using a BD FACSAria Cell Sorter. Cells were incubated overnight in R10 supplemented with 100 U/ml of IL-2, and the day after cells were infected with 350,000 TCID$_{50}$ of the NL4.3 viral strain by spinoculation (2 h, 1200 g at 37 °C). Cells were thoroughly washed and cultured in 96-well plates with R10 containing 40 U/ml IL-2 for the next 3 days. On days 0 and 3 cells were collected and subjected to flow cytometry and total RNA extraction. Upregulation of CD20 expression was identified by flow cytometry using the antibodies in 100 μl CD3 (PE-Cy7, 2.5 μl, BD 557851), CD4 (AF700, 1.25 μl, BD 557922), and CD20 (BV785, 2.5 μl, Biolegend 302356). Levels of CD20-mRNA were relatively quantified using the β-actin reference gene (PerfeCTa® qPCR FastMix II®, ROX, Quantabio) and applying the $2^{-\Delta\Delta CT}$ method.

**Quantification of cell death after Rituximab treatment**. PBMCs from uninfected donors and ART-suppressed HIV-infected individuals were cultured with R10 in the presence of 10 μg/ml Rituximab or the control antibody Palivizumab for 48 h. Subsequently, cells were processed for flow cytometry and stained with violet viability dye (Invitrogen, L34963), Annexin V (PE, 2 μl in 100 μl, Biolegend 640907), CD3 (PE-Cy7, 2.5 μl in 100 μl, BD 557851), CD4 (AF700, 1.25 μl in 100 μl, BD 557922), and CD20, for which we used the combination of Rituximab and anti-human AlexaFluor488 (ThermoFisher, 31531). The analysis was performed using a LSRFortessa cytometer. Antibody-dependent cell-mediated cytotoxicity (ADCC) was measured as follows. Target CD4$^+$ T cells were isolated from uninfected donor PBMCs as described above. CD4$^+$ T cells were stained with the cell proliferation dye eF670 following manufacturer's instructions (Labclinics) and then incubated with 10 μg/ml of Rituximab for 30 min; cells treated with with the control antibody Palivizumab were used as a negative control. Target cells were plated in U-bottom

96-well plates (5.000 cells/well, 20 wells per condition). In parallel, NK cells were isolated from PBMCs from uninfected donors using a commercial kit (MagniSort™ Human NK cell Enrichment Kit, eBiosciences) and plated at 100.000 cells/well (ratio target:effector of 1:20). After incubation, cells were stained with 10 µg/ml Rituximab plus anti-human Alexa Fluor 488 and then analyzed using a LSRFortessa cytometer.

**Ex vivo viral reactivation of CD4+ T cells.** CD4+ T lymphocytes were isolated as described above and cultured in R10 alone or stimulated with romidepsin (40 nM), ingenol (100 nM), or PMA/ionomycin (PMA, 81 nM; ionomycin 1 µM) for 22 h in the presence of raltegravir (1 µM). Then, cells were washed and subjected to the RNA FISH-flow assay, as described above. For HIV RNA detection, we used target-specific Alexa Fluor 647-labeled probes. Surface staining was performed in 100 µl with CD3 (AF700, 1 µl, Biolegend 300423), CD95 (PE-Cy5, 10 µl, BD 559773), HLA-DR (BV711, 0.6 µl, Biolegend 307644), CD20 (BV785, 2.5 µl, Biolegend 302356), CCR7 (PE-CF594, 1 µl, BD 150503), p24 (PE, 5 µl, Beckman Coulter), CD45RO (BV605, 1.25 µl, Biolegend 304238), and CD27 (FITC, 4 µl, BD 555440). Samples were analyzed using a LSRFortessa flow cytometer.

**Quantification of HIV RNA after viral reactivation.** Total PBMCs from ART-suppressed patients were incubated for 21 h with R10 containing raltegravir (1 µM), nevirapine (100 nM), 10 µg/ml Rituximab or the control antibody Palivizumab, and the LRA romidepsin (40 nM) or panobinostat (30 nM). The LRA was then extensively washed, and cells were cultured for an additional 24 h in R10 with 10 µg/ml Rituximab or the control antibody. Controls consisting of a cell culture without the LRA and a culture with the control antibody Palivizumab were included for each sample tested. After incubation, CD4+ T cells were isolated by negative selection as described above, and intracellular RNA was extracted and quantified as described above. The fold-change reactivation was calculated by dividing the quantification of HIV RNA (molecules of HIV RNA per million CD4+ cells) produced in the presence of romidepsin or panobinostat by the levels (or quantification) of viral RNA without the addition of the LRA. The fold-change depletion of HIV RNA induced by Rituximab was calculated by dividing the levels of viral RNA induced by romidepsin or panobinostat by the HIV RNA levels after LRA and Rituximab incubation.

**Cell activation.** PBMCs from uninfected donors were cultured in R10 plus 20 U/ml IL-2 with or without the activating antibodies anti-CD3 (1 µg/ml) and anti-CD28 (2 µg/ml). Expression levels of CD20, CD69, and HLA-DR were analyzed at 0, 24, 48, 72, and 96 h. At each time point, cells were stained with 100 µl CD4 (AF700, 1.25 µl, BD 557922), CD3 (PE-Cy7, 2.5 µl, BD 557851), CD20 (FITC, 2 µl, Biolegend 302303), HLA-DR (Pe-Dazzle 594, 2.5 µl, BioLegend 307653), CD69 (PE, 2.5 µl, Biolegend 310905), and CD14 (APC, 5 µl, BD 561708), and samples were analyzed using a LSRFortessa flow cytometer.

**Statistical analyses.** Statistical analyses were performed with Prism software, version 6.0 (GraphPad). Nonparametric Mann–Whitney $U$ tests, Wilcoxon rank and ANOVA Dunn's multiple comparison tests were used as appropriate. For correlations, Spearman's correlation coefficient was calculated. A $p$ value < 0.05 was considered significant.

**Reporting summary.** Further information on research design is available in the Nature Research Reporting Summary linked to this article.

## Data availability
Data underlying figures are provided as Source Data files. All other data are available from the corresponding author upon reasonable requests.

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

## Acknowledgements

We are indebted to the Cytomics core facility of the IDIBAPS for their technical help with the Amnis® Flow cytometer. This study was supported by the American National Institutes of Health (grant R21AI118411 to M.B.), the Spanish Secretariat of Science and Innovation and FEDER funds (grant SAF2015-67334-R [MINECO/FEDER]), the Spanish "Ministerio de Economia y Competitividad, Instituto de Salud Carlos III" (ISCIII, PI17/01470), GeSIDA and the Spanish AIDS network Red Temática Cooperativa de Investigación en SIDA (RD16/0025/0007). M.B. is supported by the Miguel Servet program funded by the Spanish Health Institute Carlos III (CP17/00179). M.G. is supported by the "Pla estrategic de recerca i innovacio en salut" (PERIS), from the Catalan Government. The funders had no role in study design, data collection and analysis, the decision to publish, or preparation of the paper.

## Author contributions

M.B. designed, directed, and interpreted experiments. C.S.-P., J.G.-E., L.L.-B., A.A.-G., J.G.-R. and M.G. performed experiments, analyzed the data, and interpreted experiments. R.W., J.N., A.C., J.B., E.R., M.M., B.R., A.T., B.P., F.G., R.B., J.C. and V.F. were responsible for recruitment, specimen handling and storage, and related clinical data collection. M.B. and C.S.-P. wrote the initial paper, and all of us contributed to editing of the paper.
