## [Peer Review File · Nature Communications]

Reviewers' comments:

Reviewer #1 (Remarks to the Author):

In their manuscript entitled “Expression of CD20 after Viral Reactivation Renders HIV-reservoir Cells Susceptible to Rituximab”, the authors report a new cell subset that they identified as an HIV reservoir, and show that Rituximab is effective in depleting it. This may provide an approach for reducing the latent HIV reservoir. Yet, it is not clear whether the reduction provided by this depleting approach will be enough to exert any impact on virus rebound at the cessation of ART and towards a functional cure. Another critical issue that is missing from the paper is the relation with the Tfh cells, which may critically interact with the CD20 positive T cells.

Major comments

1. The authors show that the expression and distribution of CD20dim expression in CD4+ subsets is similar between healthy donors, ART suppressed patients, and viremic patients in Figure S1, however in Figure 1D and E, where the authors show the significant increase in expression and distribution of CD20 among TCM, TEM and TTM, they utilize only data from ART suppressed and viremic patients. Is there a particular reason for this?
2. In the legend for Figure 1, the method of statistics for panels C and D are described, however, this description is not given for the statistics used in panel E.
3. Under the results heading “Sorted CD20dim CD4+ cells have similar levels of viral DNA compared to other CD4+ T cells”, the authors state that sorting is performed to isolate total CD4+ cells, activated CD4+ T cells, resting CD4+ T cells, activated CD20dim CD4+ T cells, and resting CD20dim CD4+ T cells. However, there is data provided for HIV-DNA copies in activated and resting CD20 CD4+ T cells. How is this estimated if these cells were not isolated? It is also not clear on how the calculation for Figure 3B is done.
4. In figure 4B, the authors state that data from 2HIV-viremic and 2 ART-suppressed patients is used, however for each data set, there are many more points shown. What do these points show, and how was statistical analysis performed if there were only 2 patients per data set? Data is also similarly shown for Figure4E.
5. In figure 4C, data for both viremic and ART-suppressed individuals is combined to show correlation. It seems that all of the ART-suppressed individuals have data clustered at the lower end of the correlation graph. It seems as if the true correlation is only for viremic individuals. Also, for Figure4, there is no legend detailing what the colors orange and red signify in the graphs.
6. In Figure6A and B, why is data from healthy donors mixed with data from ART-suppressed HIV+ patients? In Figure6A, it appears that the late apoptosis level in healthy donors is much higher than ART-suppressed individuals, on administration of Rituximab. In Figure 6E, there is no legend detailing what the colors blue and white signify in the figure.

7. In TableS1, some patients are marked with 0 months on ART, however they have an ART regimen listed under their ART regimen category. What does this mean.
8. The references do not correspond with the listed work and have to be revised. Two critical papers reporting reduction of viral load after administration of rituximab (Gaufin Blood and Fukasawa Nat Med) are not cited.
9. The manuscript is very poorly written and the English would massively benefit if the manuscript is reviewed by a native English speaker. As it stands now, the story is difficult to follow.
10. The authors refers repeatedly to the CD32a as a reservoir. These results are subject to a massive controversy and the authors should refer to these other papers.

Minor comments

1. In lines 98-99, "expression of HLA-DR, CD69 or CD25". Does this mean the expression of each of these markers was found and then summed to obtain this data, or does it signify that these cells are triple positive for these markers?

Reviewer #2 (Remarks to the Author):

This is an interesting article that examines CD20 expression on CD4+ T cells and its relationship to HIV transcriptional activity and infection. The identification of markers of either transcriptionally active or latently infected cells is certainly on the forefront of HIV curative strategies, and the authors provide some interesting data to support CD20 as a potential marker of a subset of HIV RNA+ CD4+ T cells. The identification of CD20 is interesting, but there are some methodological concerns and questions remaining regarding the direct relationship between CD20 expression and HIV.

One of the overall concerns is the identification of CD4+ T cells using the flow methods presented. Of note, given recent findings that CD32 expression on rare CD4+ T cells may represent doublets (even with careful gating) or complexes associated with CD4 that express CD32, it would be necessary to include other classical B cell markers in the flow gating strategy or specifically exclude cells that express CD19, for example, from the analysis. While gating on CD3/CD4 was rigorous, showing similar results while gating out other B (or even myeloid) cell markers is of utmost importance. This would also be the case for cell sorting experiments. It would also be nice to show how CD20 expression on non CD4+ T cells fluctuates with HIV infection or transcriptionally active state. Some of the cells also do not appear dim, and seems like CD20+ or not more aptly defines how the flow gating is performed (some cells express higher levels, but seem to be included in the "dim" analyses).

) Also, more information as to how gating was determined on the populations should be provided- e.g. were samples stained with all markers except CD20 used to set up gating strategies? This is particularly helpful when applying to relatively few, dimly-expressing cells.

One of the findings that is somewhat difficult to explain with the presented data and also presents some concerns regarding data presented is the similar frequencies of CD20+ CD4+ T cells in uninfected and infected patients (a majority of patient samples were similar between groups with the exception of a few outliers). If only a few patients had higher expression patterns, why are CD4+ T cells in uninfected patients expressing CD20? One could make the argument that CD20 is more of an alternative or non classical marker of cell activation than specific for HIV infection. This finding was also observed in recent CD32 studies and initial results have been difficult to replicate. Either way, some explanation for the similar expression patterns on infected and uninfected patients would be helpful. PD1 and other markers of cell activation have also been associated with enriched HIV nucleic acid levels, and it is unclear how this one would be more beneficial given that CD20 is expressed on many more non -CD4+ T cells than on CD4+ T cells.

In figure 1. summary data is shown for healthy individuals, but would be good to show the HIV RNA and CD20 flow diagrams for uninfected individuals given similar levels in many patients on ART with healthy donors.

In Fig 3 A the DNA enrichment was modest, but the theme of the paper is to show increased RNA levels or frequency of RNA producing cells. However, it is not clear what the sequence data in Fig 3c- contributes to this manuscript. A more pertinent measure would be to investigate full-genome integrity or some surrogate for viral outgrowth in the CD20 expressing cells.

The images in Figure 4 are difficult to and the rank correlation presented is really only positive due to by 3 outliers as the rest of the population does not appear to have any direction. Furthermore, in Figure 4E, the data seem to show a large number of p24+ cells in B cell follicles than CD20+ p24+ cells in B cell follicles; although CD20+ enriched in RNA, the absolute numbers are much lower than the total numbers observed on p24 total cells which limits targeting the marker as a therapeutic strategy, even with reservoir latency reversal as below.

Data presented in figure 5 are intriguing and show an increase in CD20+ cells after ex vivo infection. But as above, this data could be consistent with CD20 being a non classical marker of activation in a minor population of CD4+ T cells.

Interestingly, there seems to be important information on the 3 clinical case descriptions that are not presented, but are some of the most interesting findings in the paper. For example, data from 3 individuals who received antiCD20 therapy show a decrease in the CD20+ CD4+ T cells, but this is wholly expected. What would be far more compelling to the argument presented in this paper, is a reduction in the overall HIV burden (DNA or RNA) in both CD4+ total and CD20+CD4+ T cells. This information will be very important in gauging the extrapolatability of the paper and would be crucial information for the manuscript. What did the flow/p24 studies and RNA studies show in these samples?

There is also a paucity of information provided in the paper summarized in Fig 7. A-C- that allows the reader to understand the importance of the impact of reservoir reactivation on CD4+ CD20+ T cells. For example, the graphs show % HIV in CD20 dim positive (in extremely low frequencies to boot) - but no data are shown how HIV RNA + CD20 negative cells respond to therapy. If RNA increases in these negative cells as well, it suggests that reactivation with anti-CD20 would still not impact a majority of the HIV reservoir. This data will be critical to interpret the reactivation experiments. Ditto for grouping the last bar graph by "responders" or "non-responders" - this is a bit contrived way of showing an impact in only a subset of samples and is a bit misleading.

Reviewer #3 (Remarks to the Author):

In this manuscript, Serra-Peinado and colleagues present evidence that the B cell marker CD20 is a marker of some HIV-infected cells in vivo, and a potential target in HIV cure strategies. First, the authors use flow cytometry to define a subpopulation of CD4 T cells bearing low levels of CD20 in HIV infected and uninfected people. They find that CD20dim CD4 T cells most frequently show CM, TM, or EM phenotypes in both infected and uninfected people. Using a FISH-Flow assay for HIV RNA, the authors find higher percentages of HIV RNA-positive cells in CD20dim than CD20- cells in both ART-treated and viremic donors. Of all HIV RNA-positive cells in blood, 18.55% and 25% are found to come from the CD20dim subset in ART-treated and viremic groups. Using FACS to sort activated or resting CD20dim or CD20- cells from 9 donors, the authors find higher levels of HIV DNA in resting CD20dim cells than in total CD4 T cells, demonstrating an enrichment for HIV DNA in the resting CD20dim population. Although most infected cells come from resting and activated CD20- cells, resting and activated CD20dim contribute a substantial proportion of the total infected population. A small number of single-copy env sequences from CD20dim and CD20- cells in each of 3 donors shows no clear sequence difference among viruses in the two subsets. Staining of CD20 combined with either CD3 or p24 is performed on lymph node tissue slices from a small number of viremic and ART-treated individuals, with colocalization between CD20 and CD3 or p24 signals that is interpreted

to reflect individual double-positive cells (CD20-positive cells that have arisen from the CD4 T cell population and, in the case of p24 staining, have become productively infected). In vitro infection of whole PBMC from 3 donors shows apparent increase in the frequency of CD20dim cells, which is associated with spreading infection that prominently involves the CD20dim population (based on FISH-Flow/p24 staining). Based on these findings, the authors attempt to deplete the CD20dim, HIV-infected CD4 T cell population in vitro using LRA treatment + Rituximab. A large amount of data is presented, with the notable finding that ART-treated individuals showing the greatest induction of HIV RNA levels in CD4 T cells upon LRA treatment also show greatest depletion of CD4 T cell-associated HIV RNA with Rituximab treatment. The authors argue that this reflects targeting of CD20 on CD20dim, HIV-infected cells, potentiated by LRA treatment, and that such a combination could target infected cells in vivo.

Overall, this reviewer's opinion is that this study contains publishable findings that may prove to be important. However, the authors appear to have assumed a biological explanation for their findings that the data do not prove, and that is likely to be incorrect. In some cases, the data fall far short of supporting the authors conclusions. Furthermore, the manuscript is fairly sloppy, with a great deal of extraneous data, multiple mislabeled figure legend panels, and graphics that are often difficult to read. Considerable additional lab work and manuscript improvement would be required to make the study suitable for Nature Communications.

Major comments:

1. The authors appear to believe that some CD20dim CD4 T cells arise from CD20- CD4 T cells that ectopically express CD20 in response to HIV infection, but the data presented do not prove this. Adhesion of non-T-cells and "trogocytosis" of non-T-cell fragments have now been suggested at multiple international meetings to be a source of non-T-cell marker expression on CD4 T cells. The authors should investigate this further with additional in vitro infection experiments using FACS-sorted CD20-negative CD4 T cells instead of whole PBMC. If a clear CD20dim CD4 T cell population arises out of pure CD20-negative CD4 T cells in samples from a small number of donors after in vitro infection, this would support the authors interpretation. It should be noted that even if the CD20 on some CD4 T cells actually comes from B cells (by trogocytosis or other mechanism), other key findings in this paper may still be valuable for the field.
2. The principal valuable finding in the paper, beyond the demonstration that some CD4 T cells in HIV-infected people are CD20dim and contain HIV, is that in vitro Rituximab treatment of LRA-stimulated PBMC from ART-treated individuals is associated in some cases with a reduction in cell-associated HIV RNA, compared to no Rituximab treatment. The authors present this as evidence that Rituximab targets the CD20 on infected CD20dim CD4 T cells and thus exposes these cells to killing (presumably by ADCC). However, the data presented do not prove that Rituximab led to depletion of LRA-stimulated HIV RNA specifically within CD20dim cells, nor do they prove that the effect was specific to anti-CD20 antibody. Proving these rigorously would require additional experiments in which CD4 T cells from ART-treated people are FACS-sorted into CD20dim and CD20-negative

subsets in the presence of antiretrovirals, stimulated with LRAs, and then treated with either Rituximab or control antibody in the presence of autologous NK cells.

3. The data presented in Figure 4 do not convince this reviewer of the existence of CD4 T cells expressing CD20 and containing p24 in lymph node. It is not clear that the white areas shown in panel A and the green/red areas shown in panel D truly indicate co-staining of individual cells. Could the white areas in A not represent areas of close apposition between CD20 and CD3 single-positive cells, or T cells bearing adherent material from B cells (Allen CD et al, Science 2007)? Could the CD20/p24 signal in D not represent B cells coated in immune complexes containing virions, or with their membranes in close apposition with FDCs bearing immune complexes/virions? These alternative explanations seem much more likely than gain of CD20 expression within infected CD4 T cells.

4. The FISH-Flow data shown in Figure 2 and the paper's description of background correction for this assay do not convince this reviewer that most FISH-Flow positive cells are real HIV-infected cells containing virus RNA. This concern could be addressed as follows:

a. Confirm that CD20dim cells contain HIV RNA using a different assay. Would suggest sorting CD20dim and CD20- CD4 T cells from a small group of additional donors and quantifying HIV RNA molecules using qRT-PCR.

b. Show negative control flow data from uninfected people as well as HIV-infected samples without the target-binding probes in the supplemental data.

c. Clarify the methods. Line 488 states that "signal in HIV-RNA channel detected in donor samples were subtracted to normalize values in HIV-infected samples." It would be helpful to include more detail of this normalization. What exactly was normalized? A mean fluorescence value? A frequency of positive events?

5. The validity of the flow cytometry phenotyping performed on CD20dim CD4 T cells is highly questionable. One reason for this is the concern that CD20 on these cells may come from transferred B cell membrane, which could also be positive for other markers (HLA-DR, CD45RO) that would confuse the analysis. In addition, according to the gating for CD20dim CD4 T cells shown in Figure 1A, it appears that most of the CD20dim population comes from the edge of the putative CD20-negative population. This makes it difficult to draw the gate objectively and accurately, in a way that is reproducible across donors. There may be nothing the authors could have done to prevent this problem, given that the level of CD20 on the CD20dim CD4 T cells is low. However, this makes it difficult to trust that the gated populations are pure. Therefore, many of the gated cells may not be "real" CD20dims. The authors should consider placing less emphasis on the activation and memory subset phenotyping results in the manuscript, and they should acknowledge the limitations of the analysis in the text. If they used FMO or isotype controls in setting the CD20 gate, this should be described in the manuscript.

6. Line graphs in Figure 5 show errors bars that are not defined, and no testing for statistical significance is shown for results in panels B, C, E, or F.

7. Much of the Discussion section consists of restatements of the Results section. Recommend removing the parts of the Discussion that restate Results and condensing the Discussion, focusing on

the significance of the findings made in the study. It is critical that the authors specifically address how CD20dim cells might be generated in the Discussion (CD20 expression by the CD4 T cells vs. trogocytosis vs. other), even if they cannot definitely settle the issue with their data.

8. The ADCC assay needs to be clarified significantly. The use of eFluor670, a commonly used proliferation tracer, as an ADCC marker needs to be explained. Flow gating of this assay should be shown.

Minor/editorial comments:

1. Frequently, the authors seem to confuse the proportion of cells expressing RNA in a given population with the total copy number of HIV RNA detected in that population. The FISH-Flow assay can quantify the proportion of HIV RNA-positive cells in a population. Quantitative RT-PCR for HIV RNA in bulk cell extracts can quantify the copy number of HIV RNA per input, but cannot measure from how many cells these copies came.

2. The authors should clarify several methodological details, as below

a. Line 473 states that CD4+ T lymphocytes were enriched from total PBMC using the MagniSort Human CD4+ T cell Enrichment kit. The depletion strategy in this kit includes an anti-CD20 antibody. Should this not have removed the CD20+ CD4+ T lymphocytes that the authors are characterizing in this study?

b. Line 475 states that RNA was converted to cDNA, followed by detection of the HIV LTR. The authors should specify the sequence and position of the reverse transcription primer and the PCR primers.

3. Labeling of symbols in line graphs should be made much clearer. For example, distinguishing the label "HIV-RNA+p24+CD20dimCD3+ T cells" from the label "HIV-RNA-p24+CD20dimCD3+ T cells" is difficult, particularly with font as small as in Fig. 5E.

4. It is difficult to be certain that CD20-dim cell frequencies are stable through cryopreservation based on the data presented in Figure 1B because the triangle (fresh) and circular (cryopreserved) samples come from different donors studied on different days.

5. The correlations between virus nuclei acid measurements and CD20dim CD4 T cells would make more sense using absolute counts of CD20dim cells, rather than percentages. Absolute CD20dim cell counts might be calculated by adjusting percentages for absolute CD4 T cell counts in the different donors.

6. Would consider replacing line 37, "but are not able to transcribe and produce viral particles" with "that is genetically intact but able to remain quiescent for prolonged periods in vivo." The way the sentence is currently written suggests that the cells can never express virus, which certainly is not what the authors intend to say.

7. Would change the wording of line 381 to read "we do not rule out that the CD20dim population might also be preferentially infected."

8. Line 119 refers to Figure S1A, but this figure has only a single panel.
9. Would suggest showing numerical P values rather than asterisks in all figures, except when absolutely too crowded to fit numbers into the figure space.
10. All figure legends should state what any errors bars in corresponding figures represent (SEM, SD, range, etc.).
11. Would suggest using “uninfected” or “HIV-negative” instead of “healthy” to designate negative control study participants.

Reviewers' comments:

Reviewer #1 (Remarks to the Author):

In their manuscript entitled "Expression of CD20 after Viral Reactivation Renders HIV-reservoir Cells Susceptible to Rituximab", the authors report a new cell subset that they identified as an HIV reservoir, and show that Rituximab is effective in depleting it. This may provide an approach for reducing the latent HIV reservoir. Yet, it is not clear whether the reduction provided by this depleting approach will be enough to exert any impact on virus rebound at the cessation of ART and towards a functional cure. Another critical issue that is missing from the paper is the relation with the Tfh cells, which may critically interact with the CD20 positive T cells.

Major comments

1. The authors show that the expression and distribution of CD20dim expression in CD4+ subsets is similar between healthy donors, ART suppressed patients, and viremic patients in Figure S1, however in Figure 1D and E, where the authors show the significant increase in expression and distribution of CD20 among TCM, TEM and TTM, they utilize only data from ART suppressed and viremic patients. Is there a particular reason for this?

First of all, the analyses represented in these graphs have been reanalyzed based on the new gating strategy. In response to the reviewer's comment, the aim of Figure S1 (now Figure S2B) was to show differences in the frequency/distribution of the CD20dim subpopulation in CD4 T cell memory subsets between cohorts of HIV-infected patients and HIV-negative donors. However, in Figure 1D and E (now Figure 1F and S2A), we were showing intra-patient differences (all HIV-positive). Nevertheless, including the HIV-negative data, which is now presented in current Figure S2B, into Figure 1F or S2A, neither alter the results nor the statistical analysis. We prefer to keep the graphs as they stand now, unless the reviewer and the editor think that including the HIV-negative data in Figure 1F and S2A would add value to the current graphs.

In addition, we agree with the reviewer that the relationship between CD20 T cells and TFH cells is an important aspect. We performed extra phenotypic analysis of CD20 T cells in blood and assessed the expression of TFH markers in some samples (PD1 and CXCR5). We observed that CD20 T cells expressed more TFH markers than their counterparts CD20- T cells. However, this percentage was relatively low (~4%) (new Figure S1C). Additionally, Figure 4A and B show the presence of T cells co-expressing CD20 within the B cell follicles, and per definition, T cells located within the B cell follicles should be TFH cells. We observed that ~3% of these follicular T cells expressed CD20 in lymph nodes. This data has been included and discussed in the new version of the manuscript (new Figure 4B).

2. In the legend for Figure 1, the method of statistics for panels C and D are described, however, this description is not given for the statistics used in panel E.

The statistical analysis used in panel E (now Figure 1F) is now included in the manuscript.

3. Under the results heading "Sorted CD20dim CD4+ cells have similar levels of viral DNA compared to other CD4+ T cells", the authors state that sorting is performed to isolate total CD4+ cells, activated CD4+ T cells, resting CD4+ T cells, activated CD20dim CD4+ T cells, and resting CD20dim CD4+ T cells. However, there is data provided for HIV-DNA copies in activated and resting CD20 CD4+ T cells. How is this estimated if these cells were not isolated? It is also not clear on how the calculation for Figure 3B is done.

Maybe it was not clear enough but we isolated total CD4+ T cells, activated CD4+ T cells, resting CD4+ T cells, activated CD20dim CD4+ T cells and resting CD20dim CD4+ T cells. We

only provided HIV-DNA data for the sorted populations as shown in Fig 3A. Regarding Figure 3B, the contribution of each subset was calculated considering the frequency of each subset (analyzed by flow cytometry) within the total CD4 compartment and their relative infection frequency. We have now included this description in the M&M section of the manuscript.

4. In figure 4B, the authors state that data from 2 HIV-viremic and 2 ART-suppressed patients is used, however for each data set, there are many more points shown. What do these points show, and how was statistical analysis performed if there were only 2 patients per data set? Data is also similarly shown for Figure 4E.

We agree with the reviewer that there was missing information to completely understand these graphs. In Figure 4B we show the number of cells co-expressing CD20 and CD3 in lymph node preparations. Each circle corresponds to the number of double positive cells in a single follicle and, in the extra follicular region each triangle corresponds to one optical field. Multiple follicles and fields were analyzed per patient. Moreover, all data was normalized to the area of the follicle or to the optical field (mm^2). Similar data has been previously reported when samples are limited (Huot, Nat Med 2017). This information has now been included in the legend of Figure 4, and we have also included additional micrographs (Figure 4A, right panels) showing clearly individual cells at higher magnification co-expressing CD20 and CD3.

In Figure 4E, we were showing cells $\text{CD20}^+\text{p24}^+$ per mm^2 present in individual B cell follicles. However, and as suggested by reviewer 3, "*the CD20/p24 signal might also represent B cells coated in immune complexes containing virions, or with their membranes in close apposition with FDCs bearing immune complexes/virions*". Although p24 staining was visualized as an intracellular staining, we cannot exclude the alternative explanation provided by reviewer 3. For this reason, in the new version of the manuscript we have excluded the quantification of $\text{CD20}^+\text{p24}^+$ cells. Alternatively, and in order to unequivocally show the presence of HIV-infected cells expressing CD20 in lymph nodes, we have performed in situ RNA hybridization of viral RNA and CD20 RNA in lymph node sections from an HIV viremic patient. New micrographs showing individual infected cells expressing CD20 have now been included in Figure 4D.

5. In figure 4C, data for both viremic and ART-suppressed individuals is combined to show correlation. It seems that all of the ART-suppressed individuals have data clustered at the lower end of the correlation graph. It seems as if the true correlation is only for viremic individuals. Also, for Figure 4, there is no legend detailing what the colors orange and red signify in the graphs.

We agree with the reviewer on what seems to be the "true correlation", which is only observed in the viremic patients. In the manuscript we speculated that CD20 may be upregulated upon HIV infection in CD3^+ cells localized within the B-cell follicles, however we did not demonstrate this association. For these reasons, we prefer to exclude this figure from the new version of the manuscript. The legend of Figure 4 has been edited, as requested by the reviewer.

6. In Figure 6A and B, why is data from healthy donors mixed with data from ART-suppressed HIV+ patients? In Figure 6A, it appears that the late apoptosis level in healthy donors is much higher than ART-suppressed individuals, on administration of Rituximab. In Figure 6E, there is no legend detailing what the colors blue and white signify in the figure.

In order to better understand if rituximab affected differently uninfected donors and ART-suppressed patients, we performed extra experiments. We added rituximab to PBMCs obtained from $n=3$ ART-treated and $n=3$ uninfected donors, and assessed the capacity of the antibody of inducing cell death in CD20-expressing cells. This new set of experiments have resulted in a very marked depletion of CD20-expressing cells (T and B cells) in both uninfected donors and ART-suppressed patients, compared to the former experiments. Consequently of this depletion, we now do not observe a significant increase in late apoptosis (instead we observed an increase in the total death of CD4^+ T CD20^{dim} and B cells). Differences between both set of experiments might have been caused by the use of different lots of the antibody. We think

that these results are even more supporting, because we demonstrate that rituximab is able to significantly decrease the number of T cells expressing CD20 in vitro. For clarity, we prefer to show only the impact of rituximab on decreasing the percentage of CD20-expressing cells (B and T), and not to show the apoptosis data (new Figure 6). Moreover, we have clarified in Figure 6E (now Figure 6D) in its corresponding legend the meaning of each color, as requested by the reviewer.

7. In TableS1, some patients are marked with 0 months on ART, however they have an ART regimen listed under their ART regimen category. What does this mean.

That was a mistake and we have now corrected Table S1 accordingly.

8. The references do not correspond with the listed work and have to be revised. Two critical papers reporting reduction of viral load after administration of rituximab (Gaufin Blood and Fukasawa Nat Med) are not cited.

All references have carefully been checked and appropriate modifications have been made. Moreover, the two studies suggested by the reviewer have been incorporated in the discussion of the manuscript. We thank the reviewer for his/her input.

9. The manuscript is very poorly written and the English would massively benefit if the manuscript is reviewed by a native English speaker. As it stands now, the story is difficult to follow.

The manuscript has now been edited by a professional native English speaker.

10. The authors refers repeatedly to the CD32a as a reservoir. These results are subject to a massive controversy and the authors should refer to these other papers.

As suggested by the reviewer, other CD32 papers have now been included in the discussion of the manuscript.

Minor comments

1. In lines 98-99, "expression of HLA-DR, CD69 or CD25". Does this mean the expression of each of these markers was found and then summed to obtain this data, or does it signify that these cells are triple positive for these markers?

"Expression of HLA-DR, CD69 or CD25" means that we take into account cells expressing HLA-DR or CD69 or CD25 or any combination between all three markers (we used antibodies labelled with the same fluorochrome (PE))

Reviewer #2 (Remarks to the Author):

This is an interesting article that examines CD20 expression on CD4+ T cells and its relationship to HIV transcriptional activity and infection. The identification of markers of either transcriptionally active or latently infected cells is certainly on the forefront of HIV curative strategies, and the authors provide some interesting data to support CD20 as a potential marker of a subset of HIV RNA+ CD4+ T cells. The identification of CD20 is interesting, but there are some methodological concerns and questions remaining regarding the direct relationship between CD20 expression and HIV.

One of the overall concerns is the identification of CD4+ T cells using the flow methods presented. Of note, given recent findings that CD32 expression on rare CD4+ T cells may represent doublets (even with careful gating) or complexes associated with CD4 that express CD32, it would be necessary to include other classical B cell markers in the flow gating strategy or specifically exclude cells that express CD19, for example, from the analysis. While gating

on CD3/CD4 was rigorous, showing similar results while gating out other B (or even myeloid) cell markers is of utmost importance. This would also be the case for cell sorting experiments. It would also be nice to show how CD20 expression on non CD4+ T cells fluctuates with HIV infection or transcriptionally active state. Some of the cells also do not appear dim, and seems like CD20+ or not more aptly defines how the flow gating is performed (some cells express higher levels, but seem to be included in the "dim" analyses.) Also, more information as to how gating was determined on the populations should be provided- e.g. were samples stained with all markers except CD20 used to set up gating strategies? This is particularly helpful when applying to relatively few, dimly-expressing cells.

We agree with the reviewer that this is a critical aspect. Initially, gating strategy of CD20 dim cells was set up using an FMO control (to delineate the left limit of the gate) and the expression levels observed in B cells (to delineate the right limit of the gate). This gating strategy has been incorporated in the new version of the manuscript (Figure 1A). Moreover, and as suggested by the reviewer, we have performed the same analysis in some samples including the CD19 marker. We now show in Figure S1A that results do not change when B cells (CD20⁺ and CD19⁺) are excluded from the analysis.

In addition, and in order to unequivocally show that CD4 T cells express dim levels of CD20 and it is not the result of cell conjugates (as previously identified for the CD32 marker), we have performed a set of new experiments using imaging flow cytometry (Amnis®). This technology allows the direct visualization of cells in a microscopy coupled to a flow cytometer machine. As shown in the new Figure 1B, T cells (expressing CD3 and CD4) can also express dim levels of CD20. However, and in agreement with the published CD32 data, cells expressing high levels of CD20 corresponded to cell doublets (conjugates of T and B cells, which were excluded from the analyses) (new Figure S1B). Furthermore, we noticed that T cells expressing very dim levels of CD20 (not included in our new analysis) were the result of a punctuate staining localized in the membrane of the cells. Whereas this staining pattern could be compatible with trogocytosis events, this phenomenon was also observed for all individual antibodies added to the panel. Thus, unspecific conglomerates of antibodies in the membrane of some cells seems the most likely explanation. All this information has been included in the new version of the manuscript, and as already mentioned, the new analyses have been restricted to avoid these contaminations. Of note, after obtaining these results we applied a very stringent gating strategy (Figure 1A). New analysis of the existing data, and the addition of new samples (n= 21 ART-treated and n=20 Viremics) show that viremic HIV-infected patients have higher percentages of CD20 expression in CD4 T cells compared to ART-treated and healthy donors. Moreover, we observed that the frequency of CD20 expression on T cells correlated with the time on suppressive ART and with CD4 T cell counts (new Figure 1D). This is an important finding because it demonstrates that HIV replication in vivo has an impact on the expression of CD20 in CD4 T cells. All this information has now been included in the new version of the manuscript.

Also, as suggested by the reviewer, we have checked CD20 expression on non-CD4 T cells. After gating on the population CD3⁺CD4⁻ (enriched in CD8 T cells), we determined the expression of CD20dim on these cells, and levels were higher than those observed in CD4 T cells. This phenomenon has been previously reported in Multiple Sclerosis patients (Palanichamy et al.,2014 J Immunol). In contrast to CD4 T cells, no significant differences were observed between the different cohorts of individuals (graph below). We prefer not to include this information in the manuscript, as it is out of the scope of the study.

One of the findings that is somewhat difficult to explain with the presented data and also presents some concerns regarding data presented is the similar frequencies of CD20+ CD4+ T cells in uninfected and infected patients (a majority of patient samples were similar between groups with the exception of a few outliers). If only a few patients had higher expression patterns, why are CD4+ T cells in uninfected patients expressing CD20? One could make the argument that CD20 is more of an alternative or non-classical marker of cell activation than specific for HIV infection. This finding was also observed in recent CD32 studies and initial results have been difficult to replicate. Either way, some explanation for the similar expression patterns on infected and uninfected patients would be helpful. PD1 and other markers of cell activation have also been associated with enriched HIV nucleic acid levels, and it is unclear how this one would be more beneficial given that CD20 is expressed on many more non-CD4+ T cells than on CD4+ T cells.

As explained above, the new stringent analysis of the flow data and the addition of new samples to our different cohorts show now that frequencies of CD20dim CD4 T cells are higher in viremic HIV-infected patients compared to uninfected controls.

We agree with the reviewer that CD20 could act as an activation marker. Consistent with the ex vivo infection experiments, where an increase of CD20 was observed over time in cells with an intact CD4 expression (early infection), we hypothesize that CD20 is an activation cell marker, most likely a non-classical activation marker, which is transiently upregulated during the early stages of the infection. This could explain why not all infected cells express CD20. In addition, CD20 is upregulated after viral reactivation (new Figure 8). Said that, targeting CD20, and no other activation markers, would have some fundamental advantages, such as CD20 is not ubiquitously expressed in all T cells, and anti-CD20 depleting antibodies in vivo have proved to be relatively safe, a significant difference with current anti-PD1 antibodies. Moreover, anti-CD20 antibodies target a cellular protein, therefore the generation of resistance mutations would be highly improbable, in contrast to results observed with current anti-HIV monoclonal antibodies. Altogether, we believe that CD20 is a non-classical activation marker that is transiently expressed early upon HIV infection and after viral reactivation. Importantly, this marker can be targeted with anti-CD20 therapies. This discussion has been added to the new version of the manuscript.

In figure 1. summary data is shown for healthy individuals, but would be good to show the HIV RNA and CD20 flow diagrams for uninfected individuals given similar levels in many patients on ART with healthy donors.

We assume that the reviewer refers to Figure 2. This data has been incorporated in the new version of the manuscript (new Figure 2A).

In Fig 3 A the DNA enrichment was modest, but the theme of the paper is to show increased RNA levels or frequency of RNA producing cells. However, it is not clear what the sequence data in Fig 3c- contributes to this manuscript. A more pertinent measure would be to investigate full-genome integrity or some surrogate for viral outgrowth in the CD20 expressing cells.

We fully agree with the reviewer that a more pertinent measure would be to show the full genome sequencing or the viral outgrowth assay. However, the transient expression of CD20 in HIV-infected cells (or viral-reactivated cells) impedes the correct evaluation of the viral landscape present in CD20dim cells, or CD4 T cells becoming CD20. Moreover, due to the low frequency of these cells, we would need to initiate the cell sorting with large quantities of PBMCs in order to obtain enough cells to run the viral outgrowth assay. Note that after cell sorting (starting material 100M PBMCs) we usually obtain 5.000-10.000 highly pure CD4 T CD20 dim cells. Based on the infection levels of these cells (around 0.001, as estimated by the total HIV-DNA quantification), and the average IUPM of ART-suppressed patients of 1 (Infections Units per Million Cells), we would need to initiate the sorting with a minimum of 10.000M PBMCs to obtain a reasonable quantity of cells to perform the viral outgrowth assay. For all these reasons, we finally performed sequencing of the envelope region, which is a good marker for viral quasispecies diversity, and it is an easily amplified target by PCR. Our results show no phylogenetic differences between CD20dim and total CD4 T cells, suggesting that viruses from CD20 cells are not intrinsically different than virus observed in total CD4 T cells.

The images in Figure 4 are difficult to and the rank correlation presented is really only positive due to by 3 outliers as the rest of the population does not appear to have any direction. Furthermore, in Figure 4E, the data seem to show a large number of p24+ cells in B cell follicles than CD20+ p24+ cells in B cell follicles; although CD20+ enriched in RNA, the absolute numbers are much lower than the total numbers observed on p24 total cells which limits targeting the marker as a therapeutic strategy, even with reservoir latency reversal as below.

As pointed out by the reviewer, the correlation in Figure 4C is only driven by the data corresponding to the viremic patients. Thus, we have removed this correlation from the Figure.

Moreover, due to significant concerns also raised by reviewer 3 regarding Figure 4E "*the CD20/p24 signal might also represent B cells coated in immune complexes containing virions, or with their membranes in close apposition with FDCs bearing immune complexes/virions*", and therefore, the difficulty of accurately quantify double positive cells, we have removed this graph from the manuscript. Micrographs will remain in the figure for an illustrative purpose. Instead, we have performed RNA in situ hybridization in lymph node sections of an HIV-viremic patient. This new data added to Figure 4D unequivocally shows the expression of CD20 in HIV-RNA⁺ cells. Around 80% of the RNA⁺ cells also expressed CD20 in this particular case. Because this experiment was performed in a single patient with the purpose of showing the expression of CD20 in infected cells from lymph nodes, we prefer not to show this quantification.

Moreover, we believe that even in the case that p24⁺CD20⁻ cells would outperform in numbers the population of p24⁺CD20⁺, after latency reversal we expect the transient expression of CD20 in viral-reactivated cells. Therefore, targeting this population might have a measurable impact on the number of reservoir cells.

Data presented in figure 5 are intriguing and show an increase in CD20+ cells after ex vivo infection. But as above, this data could be consistent with CD20 being a non classical marker of activation in a minor population of CD4+ T cells.

We agree with the reviewer. As explained above, we believe that CD20 would still be an interesting marker to target recently infected cells or viral reactivated cells.

Interestingly, there seems to be important information on the 3 clinical case descriptions that are not presented, but are some of the most interesting findings in the paper. For example, data from 3 individuals who received antiCD20 therapy show a decrease in the CD20+ CD4+ T cells, but this is wholly expected. What would be far more compelling to the argument presented in this paper, is a reduction in the overall HIV burden (DNA or RNA) in both CD4+ total and CD20+CD4+ T cells, This information will be very important in gauging the extrapolatability of the paper and would be crucial information for the manuscript. What did the flow/p24 studies and RNA studies show in these samples?

We thank the reviewer for bringing up this important aspect of the manuscript. In the first version of the manuscript data presented corresponded to 1 patient treated with 2 doses of RTX and with 3 longitudinal samples. We have now included samples from a second patient (new Figure 7). Overall, we show that CD4 CD20dim cells are cleared after RTX administration in both patients. As suggested by the reviewer, we performed HIV-RNA quantification in CD4 T cells. As CD20dim cells were eliminated by RTX, we decided not to perform the RNA-FISH in CD4T CD20dim because no CD20 cells would be visualized. In both patients, we observed a significant decrease of the HIV-RNA after RTX treatment. Both cases are introduced in the new version of the manuscript and a discussion for each individual case is provided.

There is also a paucity of information provided in the paper summarized in Fig 7. A-C- that allows the reader to understand the importance of the impact of reservoir reactivation on CD4+ CD20+ T cells. For example, the graphs show % HIV in CD20 dim positive (in extremely low frequencies to boot) - but no data are shown how HIV RNA + CD20 negative cells respond to therapy. If RNA increases in these negative cells as well, it suggests that reactivation with anti-CD20 would still not impact a majority of the HIV reservoir. This data will be critical to

interpret the reactivation experiments. Ditto for grouping the last bar graph by "responders" or "non-responders"- this is a bit contrived way of showing an impact in only a subset of samples and is a bit misleading.

As suggested by the reviewer, we have added new data showing the response to therapy of CD20 negative cells. New Figure 8A shows that percentages of HIV-RNA⁺ cells are significantly higher in CD20dim cells compared to CD20 negative cells. Moreover, if the reviewer and the editor agree, we would prefer to keep the last graph showing the differences between responders and non-responders because we want to emphasize that only those patients who reactivate the virus would benefit of the rituximab treatment.

Reviewer #3 (Remarks to the Author):

In this manuscript, Serra-Peinado and colleagues present evidence that the B cell marker CD20 is a marker of some HIV-infected cells in vivo, and a potential target in HIV cure strategies. First, the authors use flow cytometry to define a subpopulation of CD4 T cells bearing low levels of CD20 in HIV infected and uninfected people. They find that CD20dim CD4 T cells most frequently show CM, TM, or EM phenotypes in both infected and uninfected people. Using a FISH-Flow assay for HIV RNA, the authors find higher percentages of HIV RNA-positive cells in CD20dim than CD20⁻ cells in both ART-treated and viremic donors. Of all HIV RNA-positive cells in blood, 18.55% and 25% are found to come from the CD20dim subset in ART-treated and viremic groups. Using FACS to sort activated or resting CD20dim or CD20⁻ cells from 9 donors, the authors find higher levels of HIV DNA in resting CD20dim cells than in total CD4 T cells, demonstrating an enrichment for HIV DNA in the resting CD20dim population. Although most infected cells come from resting and activated CD20⁻ cells, resting and activated CD20dim contribute a substantial proportion of the total infected population. A small number of single-copy env sequences from CD20dim and CD20⁻ cells in each of 3 donors shows no clear sequence difference among viruses in the two subsets. Staining of CD20 combined with either CD3 or p24 is performed on lymph node tissue slices from a small number of viremic and ART-treated individuals, with colocalization between CD20 and CD3 or p24 signals that is interpreted to reflect individual double-positive cells (CD20-positive cells that have arisen from the CD4 T cell population and, in the case of p24 staining, have become productively infected). In vitro infection of whole PBMC from 3 donors shows apparent increase in the frequency of CD20dim cells, which is associated with spreading infection that prominently involves the CD20dim population (based on FISH-Flow/p24 staining). Based on these findings, the authors attempt to deplete the CD20dim, HIV-infected CD4 T cell population in vitro using LRA treatment + Rituximab. A large amount of data is presented, with the notable finding that ART-treated individuals showing the greatest induction of HIV RNA levels in CD4 T cells upon LRA treatment also show greatest depletion of CD4 T cell-associated HIV RNA with Rituximab treatment. The authors argue that this reflects targeting of CD20 on CD20dim, HIV-infected cells, potentiated by LRA treatment, and that such a combination could target infected cells in vivo.

Overall, this reviewer's opinion is that this study contains publishable findings that may prove to be important. However, the authors appear to have assumed a biological explanation for their findings that the data do not prove, and that is likely to be incorrect. In some cases, the data fall far short of supporting the authors conclusions. Furthermore, the manuscript is fairly sloppy, with a great deal of extraneous data, multiple mislabeled figure legend panels, and graphics that are often difficult to read. Considerable additional lab work and manuscript improvement would be required to make the study suitable for Nature Communications.

Major comments:

1. The authors appear to believe that some CD20dim CD4 T cells arise from CD20⁻ CD4 T cells that ectopically express CD20 in response to HIV infection, but the data presented do not prove this. Adhesion of non-T-cells and "troglodytosis" of non-T-cell fragments have now been suggested at multiple international meetings to be a source of non-T-cell marker expression on CD4 T cells. The authors should investigate this further with additional in vitro infection

experiments using FACS-sorted CD20-negative CD4 T cells instead of whole PBMC. If a clear CD20dim CD4 T cell population arises out of pure CD20-negative CD4 T cells in samples from a small number of donors after in vitro infection, this would support the authors interpretation. It should be noted that even if the CD20 on some CD4 T cells actually comes from B cells (by trogocytosis or other mechanism), other key findings in this paper may still be valuable for the field.

We agree with the reviewer that in the first version of the manuscript we did not prove that CD20 was directly expressed in CD4 T cells. Now, we have performed extra experiments to unequivocally show that some CD4 T cells express CD20 dimly. We have performed a new set of experiments using imaging flow cytometer (Amnis®). This technology allows the direct visualization of cells. As shown in the new Figure 1B, CD4 T cells may express dim levels of CD20. However, and in agreement with the published CD32 data, cells expressing high levels of CD20 corresponded to cell doublets (new Figure S1B). Moreover, we now show that the addition of the CD19 marker, and the consequently exclusion of B cells, does not alter the frequency of CD4 T CD20dim cells (new Figure S1A). Furthermore, we analyze by qPCR the expression levels of CD20 in the population of CD4⁺ T CD20dim after cell sorting. We observed that this population expressed more CD20 levels than the population CD4⁺ T CD20-negative (new Figure S1D). All those assays consistently demonstrate the existence of CD4 T cells expressing the CD20 marker.

2. The principal valuable finding in the paper, beyond the demonstration that some CD4 T cells in HIV-infected people are CD20dim and contain HIV, is that in vitro Rituximab treatment of LRA-stimulated PBMC from ART-treated individuals is associated in some cases with a reduction in cell-associated HIV RNA, compared to no Rituximab treatment. The authors present this as evidence that Rituximab targets the CD20 on infected CD20dim CD4 T cells and thus exposes these cells to killing (presumably by ADCC). However, the data presented do not prove that Rituximab led to depletion of LRA-stimulated HIV RNA specifically within CD20dim cells, nor do they prove that the effect was specific to anti-CD20 antibody. Proving these rigorously would require additional experiments in which CD4 T cells from ART-treated people are FACS-sorted into CD20dim and CD20-negative subsets in the presence of antiretrovirals, stimulated with LRAs, and then treated with either Rituximab or control antibody in the presence of autologous NK cells.

Following the reviewer's recommendations, we sorted CD4 T CD20negative and CD20positive cells. However, highly pure CD20dim cells recovered after cell sorting were very few (between 5.000 and 10.000 from 100M PBMCs) and insufficient to perform the proposed experiment. Instead, we proved that the effect was specific to rituximab since a control antibody was included in the negative control (Palivizumab). We first observed that CD20 expression (measured by qPCR) was significantly increased in CD4 T cells after rituximab treatment (more likely as a result of receptor engagement) (Figure A, below). Moreover, this change in CD20 expression significantly correlated with depletion of HIV-RNA⁺ cells after rituximab administration (Figure B, below). This result indicates that rituximab was more effective at depleting HIV-reactivated cells in samples with higher CD20 expression levels. We believe that these findings prove the specificity of rituximab in targeting and depleting HIV-expressing cells. We prefer to keep these two graphs at the disposition of the reviewers only. The addition of the control antibody Palivizumab has been included in the M&M section of the manuscript.

A

B

3. The data presented in Figure 4 do not convince this reviewer of the existence of CD4 T cells expressing CD20 and containing p24 in lymph node. It is not clear that the white areas shown in panel A and the green/red areas shown in panel D truly indicate co-staining of individual cells. Could the white areas in A not represent areas of close apposition between CD20 and CD3 single-positive cells, or T cells bearing adherent material from B cells (Allen CD et al, Science 2007)? Could the CD20/p24 signal in D not represent B cells coated in immune complexes containing virions, or with their membranes in close apposition with FDCs bearing immune complexes/virions? These alternative explanations seem much more likely than gain of CD20 expression within infected CD4 T cells.

New micrographs have been added to Figure 4A. Now, we show individual CD3⁺ cells expressing also the CD20 receptor. Moreover, regarding Figure 4C we agree with the reviewer that the data presented in the first version of the manuscript might have several explanations. Thus, due to the difficulty of accurately quantify p24⁺ CD3⁺ double positive cells, we have removed this quantification from the Figure. Micrographs will remain in the figure for an illustrative purpose. Instead, we have performed RNA in situ hybridization in lymph node sections of an HIV-viremic patient. This new data added to Figure 4D unequivocally shows the expression of CD20-RNA in HIV-RNA⁺ cells. Because this experiment was performed in a single patient with the purpose of showing the expression of CD20 in infected cells in lymph nodes, we prefer not to show the cell quantification.

4. The FISH-Flow data shown in Figure 2 and the paper's description of background correction for this assay do not convince this reviewer that most FISH-Flow positive cells are real HIV-infected cells containing virus RNA. This concern could be addressed as follows:

- Confirm that CD20dim cells contain HIV RNA using a different assay. Would suggest sorting CD20dim and CD20⁻ CD4 T cells from a small group of additional donors and quantifying HIV RNA molecules using qRT-PCR.
- Show negative control flow data from uninfected people as well as HIV-infected samples without the target-binding probes in the supplemental data.
- Clarify the methods. Line 488 states that "signal in HIV-RNA channel detected in donor samples were subtracted to normalize values in HIV-infected samples." It would be helpful to include more detail of this normalization. What exactly was normalized? A mean fluorescence value? A frequency of positive events?

As suggested by the reviewer, we sorted CD20⁺ and CD20⁻ cells and performed HIV-RNA quantification using conventional qPCR. After performing 3 different sortings, we were only able to detect HIV-RNA in the CD20⁺ population of one patient (due to the low frequency of these cells). In this particular patient, HIV-RNA was higher in the CD20⁺ compared to the CD20⁻ population (Figure A below). Additionally, and in order to show the specificity of the RNA-flow technique, we sorted HIV-RNA⁺ and HIV-RNA⁻ cells and performed HIV-DNA quantification of 6 additional samples. We observed that the HIV-RNA⁺ fraction contained a 100 fold more HIV-DNA than cells not expressing HIV (Figure B below). Additionally, we have included in Figure 2 the RNA/flow data within the CD20dim subpopulation corresponding to an HIV-negative donor. Finally, as requested by the reviewer, a more detailed explanation of the normalization method has been added to the M&M section of the manuscript.

Of note, we have previously validated the RNA-FISH assay for detection of HIV-RNA in CD4 T cell subsets from ART-treated patients (Grau-Exposito, et al, mBio 2017).

5. The validity of the flow cytometry phenotyping performed on CD20dim CD4 T cells is highly questionable. One reason for this is the concern that CD20 on these cells may come from transferred B cell membrane, which could also be positive for other markers (HLA-DR, CD45RO) that would confuse the analysis. In addition, according to the gating for CD20dim CD4 T cells shown in Figure 1A, it appears that most of the CD20dim population comes from the edge of the putative CD20-negative population. This makes it difficult to draw the gate objectively and accurately, in a way that is reproducible across donors. There may be nothing the authors could have done to prevent this problem, given that the level of CD20 on the CD20dim CD4 T cells is low. However, this makes it difficult to trust that the gated populations are pure. Therefore, many of the gated cells may not be “real” CD20dims. The authors should consider placing less emphasis on the activation and memory subset phenotyping results in the manuscript, and they should acknowledge the limitations of the analysis in the text. If they used FMO or isotype controls in setting the CD20 gate, this should be described in the manuscript.

During this revision, we have performed several experiments to accurately show the existence of CD4 T cells expressing CD20. First, we have performed the analysis including the CD19 marker. We show in Figure S1A that percentages of CD4 T cells expressing CD20 do not vary when we exclude B cells using the two markers CD20 and CD19, thus excluding the possibility that CD20dim were cell doublets of T-B cells. Moreover, we have run the samples in an Amnis® imaging flow cytometer. We now show the unequivocal low expression of CD20 in CD4 T cells (new Figure 1A). Also, as previously reported for the CD32 marker, CD4 T cells showing high levels of CD20 resulted to be cell-to-cell conjugates (new Figure S1B). Additionally, we used an FMO control to set up the lower limit of the gate, and the expression levels observed in B cells as the upper limit of the gate. Consequently, we have applied a more stringent gating strategy (Figure 1A) and re-analyzed all the data. We hope that all these clarifications and modifications are convincing and strongly support our findings.

6. Line graphs in Figure 5 show errors bars that are not defined, and no testing for statistical significance is shown for results in panels B, C, E, or F.

Changes have been made accordingly and corresponding statistical analyses have been incorporated in the results section of the manuscript.

7. Much of the Discussion section consists of restatements of the Results section. Recommend removing the parts of the Discussion that restate Results and condensing the Discussion, focusing on the significance of the findings made in the study. It is critical that the authors specifically address how CD20dim cells might be generated in the Discussion (CD20 expression

by the CD4 T cells vs. trogocytosis vs. other), even if they cannot definitely settle the issue with their data.

We have now modified the discussion section as suggested by the reviewer.

8. The ADCC assay needs to be clarified significantly. The use of eFluor670, a commonly used proliferation tracer, as an ADCC marker needs to be explained. Flow gating of this assay should be shown.

We have based our ADCC analysis on those previously reported by Gomez-Roman et al 2006. The authors used a proliferation dye as a viability dye for the ADCC analysis. Several other authors have reported the use of proliferation markers to asses ADCC activity. The rationale behind this strategy is that dying cells lose their membrane integrity and consequently, the "proliferation" marker. Moreover, the length of the assay, approximately 20 hours, precludes the use of the dye as a proliferation marker (Note that proliferation assays are routinely performed during 7 days). We have set up this ADCC assay in the lab using the "proliferation" marker eFluor670 for other studies. As shown in the figure below, this assay accurately quantifies the % of dead cells based on the marker eFluor670 (T: Target cells coated with gp120 protein) after the addition of plasma from a chronic untreated HIV-infected patient and Effector cells (E: PBMCs). As requested by the reviewer, flow gating for all 6 individuals is now shown in Figure S6C.

Minor/editorial comments:

1. Frequently, the authors seem to confuse the proportion of cells expressing RNA in a given population with the total copy number of HIV RNA detected in that population. The FISH-Flow assay can quantify the proportion of HIV RNA-positive cells in a population. Quantitative RT-PCR for HIV RNA in bulk cell extracts can quantify the copy number of HIV RNA per input, but cannot measure from how many cells these copies came.

We have carefully revised the entire manuscript and corrected the mistaken sentences.

2. The authors should clarify several methodological details, as below
a. Line 473 states that CD4+ T lymphocytes were enriched from total PBMC using the MagniSort Human CD4+ T cell Enrichment kit. The depletion strategy in this kit includes an anti-CD20 antibody. Should this not have removed the CD20+ CD4+ T lymphocytes that the authors are characterizing in this study?

The editor is correct, the MagniSort kit includes an anti-CD20 antibody. However, only cells with high levels of CD20 (B cells) are correctly depleted. In the figure below we show the percentage of CD20dim within the CD4 T cells before and after cell isolation (Figure below).

b. Line 475 states that RNA was converted to cDNA, followed by detection of the HIV LTR. The authors should specify the sequence and position of the reverse transcription primer and the PCR primers.

As requested, we have included the details for the cDNA and PCR conditions used in this study in the corresponding M&M section.

3. Labeling of symbols in line graphs should be made much clearer. For example, distinguishing the label "HIV-RNA+p24+CD20dimCD3+ T cells" from the label "HIV-RNA-p24+CD20dimCD3+ T cells" is difficult, particularly with font as small as in Fig. 5E.

We have now changed the labels in Figure 5 to make clearer all symbols.

4. It is difficult to be certain that CD20-dim cell frequencies are stable through cryopreservation based on the data presented in Figure 1B because the triangle (fresh) and circular (cryopreserved) samples come from different donors studied on different days.

We agree with the editor about the limitation of quantifying CD20 in fresh and cryopreserved cells that were obtained from different donors. Due to the low number of fresh samples included, and the difficulty to make a solid conclusion, we have now removed the data corresponding to fresh patients. Additionally, we have added more samples to our existing cohorts of ART-suppressed and viremic patients.

5. The correlations between virus nuclei acid measurements and CD20dim CD4 T cells would make more sense using absolute counts of CD20dim cells, rather than percentages. Absolute CD20dim cell counts might be calculated by adjusting percentages for absolute CD4 T cell counts in the different donors.

We have performed the calculations requested by the editor. However, the low numbers of CD4 T cell counts found in viremic patient bias the results; therefore viremic patients have lower number of absolute CD4 T CD20dim cells, and consequently, absolute numbers inversely correlated with viral HIV-RNA and figure below). Because of this reason, we prefer to show the percentage of CD20dim instead of the absolute counts in the manuscript.

6. Would consider replacing line 37, "but are not able to transcribe and produce viral particles" with "that is genetically intact but able to remain quiescent for prolonged periods in vivo." The way the sentence is currently written suggests that the cells can never express virus, which certainly is not what the authors intend to say.

We have accordingly changed this line.

7. Would change the wording of line 381 to read "we do not rule out that the CD20dim population might also be preferentially infected."

We have accordingly changed this line.

8. Line 119 refers to Figure S1A, but this figure has only a single panel.

It has been corrected

9. Would suggest showing numerical P values rather than asterisks in all figures, except when absolutely too crowded to fit numbers into the figure space.

We have now included numerical P values when possible.

10. All figure legends should state what any errors bars in corresponding figures represent (SEM, SD, range, etc.).

Error bars have been specified in every figure legend.

11. Would suggest using "uninfected" or "HIV-negative" instead of "healthy" to designate negative control study participants.

We have accordingly changed this label

Reviewers' comments:

Reviewer #1 (Remarks to the Author):

The authors have appropriately addressed the comments that were previously provided. A few minor comments are as follows:

1. In the introduction, for the sentence "Under pathological condition,..." from lines 67 to 70, there is only one reference provided. This does not support the statement that most investigations have focused on patients suffering RA, as there is only one study cited.
2. In table S1, for %CD4 and viral load, the number convention used is radix to show thousands and comma used as a decimal point. This is in variation to the rest of the paper where radix is used as a decimal point, and comma is used to separate thousands. Please change the table to keep the system consistent.
3. In table S1, for patient 39, time since HIV diagnosis is 13?. Does the ? signify that time since diagnosis was not known?

S

Reviewer #2 (Remarks to the Author):

Overall this revised manuscript demonstrates much improvement over the prior version, especially with the respect to showing that CD4+ CD20dim cells are real.

What happens to CD20 expression on CD8 cells during the various stimulation and reactivation experiments shown? This data may be important in deciphering whether or not CD20 is simply a T cell activation marker (which is certainly possible versus more viral specific response)- see comments on various data presented by figure below:

Fig 1: Showing a positive control for CD19 in the Amnis experiments would be more convincing- to show that CD19 is indeed visualized in B cell controls.

Fig 2. It would be best to show HIV RNA cells by flow in CD20 negative cells rather than total cells- I realize that from these experiments RNA was lower, but would make more logical sense in the figure

to show these flow plots from the same example patient in the upper portion of the figure. Also, Panel B data seem to be replicated in panel C- and can certainly be removed.

Fig 3. Panel A should show CD4+ Total cells in both activated and resting subsets as shown for the CD20- and CD20 dim cells.

Fig 7. Changes in RNA should be shown in log scale rather than linear scale as would more appropriately show relevant magnitude differences.

Fig 8. It seems that cell stimulation alone may dramatically increase CD3+ CD20dim frequencies. It would be important to show how these drugs impact CD20dim T cells in uninfected cells. One gets the sense from the figure that CD20dim cells would increase dramatically regardless of HIV infection in the setting of these stimuli. If this is the case, it would support the argument that CD20 is an activation response rather than a viral specific response. Also, it is certainly possible that the RNA fraction increased because the number of CD20dim cells increased as well- if expanding a cell population, this would also expand the nucleic acid production as well.

Overall, it seems that by the data provided, that CD20dim cells are correlated with activation (despite slightly earlier kinetics?) and most relevant in viremic individuals. Is there any idea if this is just from global T cell activation (which seems supported by the evidence) or some viral specific process? - further discussion would be beneficial.

Reviewer #3 (Remarks to the Author):

The manuscript by Serra-Peinado et al. describes interesting findings and is much improved on revision. Recommend publication after the authors address the following issues.

1. Recommend additional experiments where CD20-negative CD4 T cells are sorted from uninfected donors, infected with HIV in vitro, and then checked for CD20 surface expression and mRNA. It remains difficult for this reviewer to accept that the CD20 detected on CD4 T cells in this study is expressed from within the CD4 T cells. Figure 5 reports “upregulation of CD20 expression,” but experiments use whole PBMC rather than purified CD20-negative CD4 T cells. The CD20 in these

experiments could be transferred from other cells. If performing these experiments is infeasible, the manuscript can acknowledge that the source of the CD20 is not clear. This should not prevent publication.

2. Consider moving imaging flow from Fig S1B into main text Fig 1. This will immediately show the reader that lack of CD19 expression on the cells currently shown in Fig 1B is not due to poor CD19 staining, and will also show that dim CD20 on these cells is not simply background staining.

3. In Fig 2B, results from the uninfected donors are not plotted on the graph, but are used for comparison to ART and viremic donors with P values plotted above the data from these groups. This is highly unusual. All data used for generating significant P values should be shown.

4. Consider removing phylogenetic trees. Too few sequences are obtained to allow meaningful phylogenetic analysis. The inclusion of these figures will therefore largely be a distraction in the manuscript.

5. Fig 6 raises several questions.

a. It is confusing that Rituximab appears to eliminate all HIV p24+ cells in the culture. What proportion of p24+ cells are CD20dim? Shouldn't Rituximab eliminate CD20dim cells, rather than being selective for HIV-infected cells?

b. Is there a reason that p24 rather than HIV RNA FISH-Flow is used to score infection in these experiments?

c. In Fig 6C, the negative control samples are marked "medium." If a negative control mAb was used, this should be indicated in the figure and/or legend.

d. In Fig 6D left panel, axis label is %p24+ and values range from 0-0.5, but legend refers to "proportions of infected cells" rather than percentages. Does a value of 0.1 mean 1/1000 cells is positive, or 1/10 cells is positive?

6. New Fig 7 raises several questions.

a. It is not clear to this reviewer that changes in cell-associated HIV RNA shown in panels B represent a true Rituximab effect. How stable are CA-RNA levels in CD4 T cells in sequential samples from ART-treated people who have not received Rituximab? Could the findings not simply represent random variation?

b. The arrowheads for patient #53 appear to occur after day 0. Presumably, Rituximab infusion occurred before the second data points in all of the graphs on the right-hand side of this figure. Is this correct?

c. It is difficult to conclude from $n = 1$ that Rituximab caused “short-term changes in the distribution and/or differentiation of CD4 T cells” (line 479), or that a change in CD4 T cell subset composition was related to HIV cell-associated RNA results (particularly when a previous paper from this group often showed more HIV-RNA+ cells in EM than CM). The data seem highly preliminary.

7. Line 440 states that “we observed a tendency to higher proportions of CD20dim CD4+ T cells expression PD-1 and CXCR5 in blood compared to cells lacking CD20 expression,” but the underlying data in Fig S1D are very weak. Only three people are studied, and for the example shown in the gating, the X5/PD1+ population in CD20dim cells appears to consist of only two (2) events.

8. Line 490, recommend changing “Conclusively” to “In conclusion”.

Reviewer #1 (Remarks to the Author):

The authors have appropriately addressed the comments that were previously provided. A few minor comments are as follows:

1. In the introduction, for the sentence “Under pathological condition,...” from lines 67 to 70, there is only one reference provided. This does not support the statement that most investigations have focused on patients suffering RA, as there is only one study cited.

We thank the reviewer for the observation. We have added two additional references to better support the statement.

2. In table S1, for %CD4 and viral load, the number convention used is radix to show thousands and comma used as a decimal point. This is in variation to the rest of the paper where radix is used as a decimal point, and comma is used to separate thousands. Please change the table to keep the system consistent.

*We have revised and changed **Table S1** in order to maintain consistency in the supplementary material.*

3. In table S1, for patient 39, time since HIV diagnosis is 13? Does the ? signify that time since diagnosis was not known?

The question mark, “?”, in patient 39 was a typo, which has now been corrected. We have confirmed that time since diagnosis in this patient was 13 months. Thank you for the observation.

Reviewer #2 (Remarks to the Author):

Overall this revised manuscript demonstrates much improvement over the prior version, especially with the respect to showing that CD4+ CD20dim cells are real.

What happens to CD20 expression on CD8 cells during the various stimulation and reactivation experiments shown? This data may be important in deciphering whether or not CD20 is simply a T cell activation marker (which is certainly possible versus more viral specific response)- see comments on various data presented by figure below:

We agree with the reviewer on the interest of knowing if CD8 also express CD20 in response to stimulation. However, we did not include appropriate markers to identify CD8⁺ T cells in the viral infection and viral reactivation experiments. Actually, the viral reactivation assays were performed with isolated CD4⁺ T cells. In order to answer the reviewer's question, we have performed extra experiments to determine the expression of CD20 in CD8⁺ T cells during reactivation assays. CD8 T cells were isolated using magnetic beads and cultured with romidepsin, ingenol or PMA/Ionomycin for 22h. Then, we assessed expression of CD20 in CD8 T cells by flow cytometry. As shown in the new **Figure 8C**, we observe that romidepsin slightly increased the expression of CD20 in CD8 T cells. However, when the data was compared to the upregulation of CD20 in CD4-infected cells, the expression of CD20 in CD8 T cells was negligible. This new data and corresponding discussion has now been included in the new version of the manuscript (results line 351, discussion line 437-448).

Fig 1. Showing a positive control for CD19 in the Amnis experiments would be more convincing to show that CD19 is indeed visualized in B cell controls.

*As suggested by the reviewer, in the new **Figure 1C** we have added three new images of B cells from the AMNIS cytometer, demonstrating that CD19 and CD20 are clearly visualized in B cells.*

Fig 2. It would be best to show HIV RNA cells by flow in CD20 negative cells rather than total cells- I realize that from these experiments RNA was lower, but would make more logical sense in the figure to show these flow plots from the same example patient in the upper portion of the figure. Also, Panel B data seem to be replicated in panel C- and can certainly be removed.

*As suggested by the reviewer, we have redesigned **Figure 2A**. We now show HIV-RNA levels in CD20^{dim} and CD20⁻ cells from the same individual. This new panel illustrates the enrichment of HIV-RNA in CD20^{dim} cells compared to CD20⁻ in all HIV-infected cohorts.*

Moreover, we agree with the reviewer that panel B and panel C contain replicated data. Accordingly, we have now removed the old panel B, and we show the same information in the combined new panel 2B.

Fig 3. Panel A should show CD4+ Total cells in both activated and resting subsets as shown for the CD20- and CD20 dim cells.

*Unfortunately we do not have the data requested by the reviewer. Data presented in **Figure 3A** corresponds to FACS-sorted cells. The cell sorting was designed to maximize the recovery of CD20 dim cells. For this reason, the separation of total CD4 T cells (resting and activated) would*

enter in conflict with the separation of CD20 dim (resting and activated). Note that, taking into account that CD20^{dim} cells represent a median of 0.7% of the total CD4⁺ cells, data from the CD20^{dim} population should be very similar to results from total CD4⁺ cells.

Fig 7. Changes in RNA should be shown in log scale rather than linear scale as would more appropriately show relevant magnitude differences.

We agree with the reviewer that RNA data in log scale would better show highly significant changes. However, we observe a consistent two and a half fold decrease in the HIV-RNA levels after rituximab treatment in two patients. In agreement with the natural variation of the RNA levels in longitudinal samples from ART-treated patients, where only 5% of patients are likely to present a 2-fold variation in HIV-RNA levels randomly (Leth S. et al AIDS 2016), these changes might be meaningful. Therefore, we would prefer to maintain the original format to highlight these changes.

Fig 8. It seems that cell stimulation alone may dramatically increase CD3+ CD20dim frequencies. It would be important to show how these drugs impact CD20dim T cells in uninfected cells. One gets the sense from the figure that CD20dim cells would increase dramatically regardless of HIV infection in the setting of these stimuli. If this is the case, it would support the argument that CD20 is an activation response rather than a viral specific response. Also, it is certainly possible that the RNA fraction increased because the number of CD20dim cells increased as well- if expanding a cell population, this would also expand the nucleic acid production as well.

We followed the reviewer's suggestion and checked the overexpression of CD20 in uninfected T cells after LRA treatment. We observed that uninfected cells slightly increased the expression of CD20 after ingenol (median 2.45%) or PMA/Iono (median 2.63%) treatment, but no changes were observed with romidepsin treatment (median 1.57%) (Figure 8C). However, the upregulation of CD20 in HIV-infected cells was more evident after LRA treatment (11.41, 12.07 and 6.66% for ingenol, PMA/iono and Romidepsin, respectively). This comparison has now been included in Figure 8C.

Note that the general activation of T cells with anti-CD3 and CD28 antibodies slightly increase the expression of CD20. However, this stimuli significantly increased the expression of CD69 and HLA-DR (Figure S5B). Additionally, we performed new experiments: FACS-isolated CD4⁺ CD20-negative T cells were infected with HIV. We observe a strong upregulation of CD20 (approx. 30%), especially in productively HIV-infected cells (new Figure S3).

Compatible with this data, we present CD20 as an unconventional activation marker that is specially overexpressed during HIV infection. However, other stimuli could also slightly increase its expression. We have included this discussion in the new version of the manuscript (line 437-448).

As also suggested by the reviewer, we believe that the expansion of the CD20 population could partially explain the increase in both CD20+ cells and cells expressing HIV-RNA. However, under our experimental settings, we believe that this argument is very unlikely because: 1) the experiment duration was of only 22h (a short period of time), and 2) the LRA should only induce proliferation in the CD20+ cells (and not in the remaining cells).

Overall, it seems that by the data provided, that CD20dim cells are correlated with activation (despite slightly earlier kinetics?) and most relevant in viremic individuals. Is there any idea if this is just from global T cell activation (which seems supported by the evidence) or some viral specific process? - further discussion would be beneficial.

*Our results indicate that CD20 expression is specially upregulated during HIV infection, but also it might slightly increase upon cell activation. We present several evidences supporting this argument: i) we demonstrated that stimulation of uninfected donors with anti-CD3 and anti-CD28 antibodies slightly promotes CD20 expression in uninfected CD4⁺ T cells (up to 4%) (**Figure 55B**); ii) LRA treatment slightly increases the expression of CD20 in HIV-negative cells (up to 5%), but the upregulation is stronger in HIV+ cells (up to 20%) (**Figure 8C**); iii) ex vivo HIV infection of FACS-sorted CD4⁺CD20-negative cells strongly upregulates CD20 (approx. 30%) (**New Figure S3**). Thus, it is tempting to speculate that viral proteins, i.e. Nef, which has been shown to modulate T cell activation (Markle TJ. et al. Future virol. 2013), may be involved in this process. More mechanistic studies are needed to finally elucidate the real contribution of the virus to CD20 upregulation. This discussion has now been added to the manuscript (line 437-448).*

Reviewer #3 (Remarks to the Author):

The manuscript by Serra-Peinado et al. describes interesting findings and is much improved on revision. Recommend publication after the authors address the following issues.

1. Recommend additional experiments where CD20-negative CD4 T cells are sorted from uninfected donors, infected with HIV in vitro, and then checked for CD20 surface expression and mRNA. It remains difficult for this reviewer to accept that the CD20 detected on CD4 T

cells in this study is expressed from within the CD4 T cells. Figure 5 reports “upregulation of CD20 expression,” but experiments use whole PBMC rather than purified CD20-negative CD4 T cells. The CD20 in these experiments could be transferred from other cells. If performing these experiments is infeasible, the manuscript can acknowledge that the source of the CD20 is not clear. This should not prevent publication.

*We agree with the reviewer on the interest of the requested experiment. Accordingly, we isolated CD4⁺ T cells that were not expressing CD20 (CD20⁻) from total PBMC of uninfected donors by FACS-sorting. Cells were then infected in vitro and the CD20 expression was measured by flow cytometry and qPCR at days 0 and 3 post-infection. The results clearly demonstrate that after HIV infection CD20⁻CD4⁺T cells are able to express CD20 de novo on their surface. In fact, approx. 30% of infected cells clearly expressed the CD20 marker in their plasma membrane. This result discards the possibility that the increase in CD20 was due to transferred cell membrane from other cell types. Moreover, we also checked by qPCR the expression of CD20. Consistently with the flow cytometry data, CD20 mRNA was higher after cell infection, these data has now been included as a new Figure (**Figure S3**).*

2. Consider moving imaging flow from Fig S1B into main text Fig 1. This will immediately show the reader that lack of CD19 expression on the cells currently shown in Fig 1B is not due to poor CD19 staining, and will also show that dim CD20 on these cells is not simply background staining.

*We agree with the reviewer’s point. Thus, we have added in **Figure 1** three new images corresponding to B cells, which clarifies the comparison between T and B cells.*

3. In Fig 2B, results from the uninfected donors are not plotted on the graph, but are used for comparison to ART and viremic donors with P values plotted above the data from these groups. This is highly unusual. All data used for generating significant P values should be shown.

*As requested by Reviewer 2, we have deleted **Figure 2B** because such data was duplicated in **Figure 2C**. Now all data used for generating the P values are shown in the new **Figure 2B**.*

4. Consider removing phylogenetic trees. Too few sequences are obtained to allow meaningful phylogenetic analysis. The inclusion of these figures will therefore largely be a distraction in the manuscript.

As requested, we have removed the phylogenetic analysis from the article.

5. Fig 6 raises several questions.

a. It is confusing that Rituximab appears to eliminate all HIV p24+ cells in the culture. What proportion of p24+ cells are CD20dim? Shouldn't Rituximab eliminate CD20dim cells, rather than being selective for HIV-infected cells?

As the reviewer mentions, probably this aspect was not clear enough in the old version of the manuscript. Figure 6D shows that only p24⁺ cells that express CD20dim are completely depleted after Rituximab treatment. In Figure 6D right panel, we show that after 3 days in culture Rituximab reduced viral infection by 40%. Collectively, these results do not support the idea that Rituximab is able to completely eliminate HIV infection.

b. Is there a reason that p24 rather than HIV RNA FISH-Flow is used to score infection in these experiments?

The HIV RNA FISH/flow assay is a time consuming and expensive assay. In this case, we were interested in showing the effect of rituximab in HIV-infected cells, and therefore, the intracellular p24 staining method, which is a much cheaper and rapid protocol, works well for this purpose.

c. In Fig 6C, the negative control samples are marked "medium." If a negative control mAb was used, this should be indicated in the figure and/or legend.

As suggested, we have changed the label of graph 6C: instead of "medium", now is labelled as "-RTX". The antibody Palivizumab was used as a negative control and this information has now been added in the figure legend.

d. In Fig 6D left panel, axis label is %p24+ and values range from 0-0.5, but legend refers to "proportions of infected cells" rather than percentages. Does a value of 0.1 mean 1/1000 cells is positive, or 1/10 cells is positive?

The value 0.1 means 1/1000, while the sentence "proportions of infected cells" was a typo, which now has been changed to "percentage of infected cells".

6. New Fig 7 raises several questions.

a. It is not clear to this reviewer that changes in cell-associated HIV RNA shown in panels B represent a true Rituximab effect. How stable are CA-RNA levels in CD4 T cells in sequential samples from ART-treated people who have not received Rituximab? Could the findings not simply represent random variation?

We agree with the reviewer that variations in the levels of HIV-RNA shown in patients treated with Rituximab are low. We observed a two and a half fold reduction in RNA levels in two patients, and based on the literature only 5% of patients might suffer a 2-fold variation in this parameter randomly (Leth S. et al. AIDS 2016). This discussion has been added to the new version of the manuscript (lines 493-495).

b. The arrowheads for patient #53 appear to occur after day 0. Presumably, Rituximab infusion occurred before the second data points in all of the graphs on the right-hand side of this figure. Is this correct?

The arrowheads in this patient were unintentionally moved. Now the arrowheads point to the right time point. The first sample was taken the same day that the patient took the first rituximab dose. We thank the reviewer for catching this mistake.

c. It is difficult to conclude from $n = 1$ that Rituximab caused “short-term changes in the distribution and/or differentiation of CD4 T cells” (line 479), or that a change in CD4 T cell subset composition was related to HIV cell-associated RNA results (particularly when a previous paper from this group often showed more HIV-RNA+ cells in EM than CM). The data seem highly preliminary.

*We agree with the reviewer, and for this reason we have removed this hypothesis from the manuscript. We revised our results, and in agreement with the reviewer, we considered that 2-3 fold decrease in intracellular HIV-RNA cannot be mathematically explained by a reduction in 10% of Central Memory T cells. Besides, we did not observe changes in the percentage of Effector Memory cells, the subset where resides a significant proportion of HIV-transcriptionally active infected cells. This discussion has been added to the manuscript (lines 495-502), and results from T_{CM} have been moved to the supplementary material (**Figure S7**).*

7. Line 440 states that “we observed a tendency to higher proportions of CD20dim CD4+ T cells expression PD-1 and CXCR5 in blood compared to cells lacking CD20 expression,” but the underlying data in Fig S1D are very weak. Only three people are studied, and for the example shown in the gating, the X5/PD1+ population in CD20dim cells appears to consist of only two (2) events.

*In order to better support our original statement, we have increased the number of samples included in **Figure S1D**. Now we observe a statistically significant difference in the proportion of cells expressing CXCR5⁺/PD-1 between CD20^{dim} and CD20⁻ T CD4 T cells.*

8. Line 490, recommend changing “Conclusively” to “In conclusion”.

Thanks for the recommendation. Done.

REVIEWERS' COMMENTS:

Reviewer #2 (Remarks to the Author):

The authors have adequately addressed my initial comments, especially with regard to showing CD20 differences on activated, uninfected CD4+ T cells. I have no further comments at this time.

Reviewer #3 (Remarks to the Author):

The authors have adequately addressed concerns about the previous version. Recommend publication.

REVIEWERS' COMMENTS:

Reviewer #2 (Remarks to the Author):

The authors have adequately addressed my initial comments, especially with regard to showing CD20 differences on activated, uninfected CD4+ T cells. I have no further comments at this time.

Thank you

Reviewer #3 (Remarks to the Author):

The authors have adequately addressed concerns about the previous version. Recommend publication.

Thank you